*Research article*

# 2  Automated riverbed composition analysis using Deep Learning
# 3  on underwater images

Alexander A. Ermilov[1], Gergely Benkő[1] and Sándor Baranya[1]
[1]Department of Hydraulic and Water Resources Engineering, Budapest University of Technology and Economics,
Budapest, 1111, Hungary
*Correspondence to:* Alexander A. Ermilov (ermilov.alexander@emk.bme.hu)
**Abstract.** The sediment of alluvial riverbeds plays a significant role in river systems both in engineering and
natural processes. However, the sediment composition can show great spatial and temporal heterogeneity, even
on river reach scale, making it difficult to representatively sample and assess. Conventional sampling methods in
such cases cannot describe well the variability of the bed surface texture due to the amount of energy and time
they would require. In this manuscript, an attempt is made to overcome this issue by introducing a novel image-
based, Deep Learning (DL) algorithm and related field measurement methodology with potential for becoming a
complementary technique for bed material samplings and significantly reducing the necessary resources. The
algorithm was trained to recognise main sediment classes in videos that were taken underwater in a large river
with mixed bed sediments, along cross-sections, using In total, 27 physical bedmaterial samples were collected
and sieved for the validation purposes. The introduced DL-based method is fast, i.e., the videos of 300-400-meter-
long sections can be analysed within minutes, with very dense spatial sampling distribution. The goodness of the
trained algorithm was evaluated i) mathematically by dividing the annotated images into test and validation sets,
and also via ii) intercomparison with other direct (sieving of physical samples) and indirect sampling methods
(wavelet-based image processing of the riverbed images), focusing on the percentages of the detected sediment
fractions. For the final evaluation, the sieving analysis of the collected physical samples were considered as the
ground truth. This meant a total of 27 measurement points, where the DL-results could be compared with the two
other methods. During data processing, outlier points, where the collected physical samples did not represent the
riverbed surface images taken there (e.g., due to bedarmour), were removed. In the remaining 14 points, the DL
algorithm produced promising results with a mean error of 4.5%. Besides, the spatial trend in the fraction changes
was also well captured along the cross-sections, based upon the visual evaluation of the footages. Furthermore,
comparison with the wavelet-based image processing justified the selection of the outlier points earlier, as its
results matched closely with the DL detections in these purely gravel-covered points and showed no sign of finer
fractions, univocally opposing the content of the physical samples. Suggestions for performing proper field
measurements are also given, furthermore, possibilities for combining the algorithm with other techniques are
highlighted, briefly showcasing the multi-purpose of underwater videos for hydromorphological assessment.
**Keywords:** riverbed texture, underwater mapping, sediment classes, Artificial Intelligence, Deep Learning,
image-based

## 35  1 Introduction

The physical composition of a riverbed plays a crucial role in fluvial hydromorphological processes, as a sort of
boundary condition in the interaction mechanisms between the flow and the solid bed. Within these processes, the

grains on the riverbed are responsible for multiple phenomena, such as flow resistance (Vanoni and Hwang, 1967; Zhou et al., 2021), stability of the riverbed (Staudt et al., 2018; Obodovskyi et al., 2020), development of bed armour (Rákóczi, 1987; Ferdowsi et al., 2017), sediment clogging (Rákóczi, 1997; Fetzer et al., 2017), fish shelter (Scheder et al., 2015), etc. Through these physical processes, the bed material composition has a determining effect on numerous river-uses, e.g., possibilities of inland waterway transport (Xiao et al., 2021), drinking water supply through bank filtration (Cui et al., 2021), or the quality of riverine habitats (Muñoz-Mas et al., 2019).. Knowledge of riverbed morphology and sediment composition (sand, gravel and cobble content) is therefore of major importance in river hydromorphology. In order to gain information about riverbed sediments, in situ field sampling methodologies are implemented.

Traditionally, bed material sampling methods are intrusive (i.e., sediment is physically extracted from the bed for follow-up analysis) and carried out via collecting the sediment grains one-by-one (areal, grid-by-number and pebble count methods, see e.g., Bunte and Abt, 2001; Guerit et al., 2018) or in a larger amount by a variety of grab samplers (volumetric methods, such as WMO, 1981; Singer, 2008). This is then followed by measuring their sizes individually on-site or transporting them to a laboratory for mass-sieving analysis (Fehr, 1987; Diplas, 1988; Bunte and Abt, 2001). These sampling procedures are time- and energy consuming, especially in large gravel and mixed bed rivers, where characteristic grain sizes can strongly vary both in time and space (Wolcott and Church, 1991; USDA, 2007), requiring a dense sampling point allocation. The same goes for critical river reaches, where significant human impact led to severe changes in the morphological state of the rivers (e.g., the Upper section of the Hungarian Danube; Török and Baranya, 2017). When assessing bed material composition on a river reach scale, experts usually try to extrapolate from the samples, and describe larger regions of the bed (even several thousand $m^2$) by data gathered in a few, several dozen points (see e.g., USDA, 2007; Haddadchi et al., 2018; Baranya et al., 2018; Sun et al., 2021). Gaining a representative amount of the sediment samples is also a critical issue. For instance, following statistical criteria such as those of Kellerhals and Bray (1971) or Adams (1979), a representative sample should weigh ten-to-hundred kg. Additionally, physical bed material sampling methods are unable to directly quantify important, hydromorphological features such as roughness or bedforms (Graham et al., 2005). Due to these constraints, surrogate approaches have recently been intensively tested to analyse the riverbed. Major examples are introduced in the rest of this section.. Unlike the conventional methods, these techniques are non-intrusive and rely on computers and other instrumentation to decrease the need of human intervention and speed up the analyses.

One group of the surrogate approaches is the acoustic methods, where an acoustic wave source (e.g., an Acoustic Doppler Current Profiler; ADCP) is pointed towards the riverbed from a moving vessel, emitting a signal. The strength and frequency of this signal is measured while it passes through the water column, reflecting back to the receiver from the sediment transported by the river, and finally from the riverbed itself. This approach is fast and larger areas can be covered relatively quickly (Grams et al., 2013). While it has already become widely used for describing sediment movement (i.e., suspended sediment, Guerrero et al., 2016; bedload, Muste et al., 2016; and indirectly flow velocity; Shields and Rigby, 2005) and channel shape (Zhang et al., 2008), it has not reached similar breakthrough for riverbed material analysis. Researchers experimented with the reflecting signal strength [dB] from the riverbed (e.g., Shields, 2010) to establish its relationship with the riverbed material. Their hypothesis

was that the absorption (and hence the reflectance) of the acoustic waves reaching the bed correlates with the type
of bed sediment.  Following initial successes, the method presented several disadvantages and limitations, hence
it could not establish itself as surrogate method for riverbed material measurements so far. For example, Shields
(2010) showed that it was necessary to apply instrument specific coefficients to convert the signal strength into
bed hardness, and these coefficients could only be derived by first validating each instrument using collected
sediment samples with corresponding ADCP data. Moreover, the method was sensitive to the bulk density of the
sediment and to bedforms. Based on his results and observations, the sediment classification could only extend to
differentiate between cohesive (clay, silt) and non-cohesive (sand, gravel) sediment patches, but gravel could not
be distinguished strongly from sand as they produced similar backscatter strengths. Buscombe et al. (2014a;
2014b) further elaborated on the topic and successfully developed a better, less limited, decision tree-based
approach. They showed that spectral analysis of the backscatter is much more effective for differentiating the
sediment types compared to the statistical analysis used by Shields. With this approach it became possible to
classify homogenous sand, gravel, and cobble patches. However, Buscombe et al. (2014a, 2014b) also emphasizes
that acoustic approaches are not capable of separating the effects of surface roughness from the effects of
bedforms, therefore the selection of an appropriate ensemble averaging window size is of great importance for
their introduced method. This size should be small enough to not include morphological signal, for which
however, the a priori analyses of riverbed elevation profiles is needed at each site. Furthermore, they suggest their
method is sensitive to and limited by high concentrations of (especially cohesive) sediment, therefore its
application to heterogeneous riverbeds would require site specific calibrations. The above-mentioned studies also
note that acoustic methods in general inherently do not allow the measurement of individual sediment grains due
to their spatial averaging nature. The detected signal strength correlates with the median grainsize of the covered
area, information about other nominal grainsizes cannot be gained.

Another group of the surrogate approaches is the application of photography (Adams, 1979; Ibbekken and
Schleyer, 1986) and later computer vision or image-processing techniques. During the last two decades, two major
subgroups emerged: one uses object- and edge detection (by finding abrupt changes in intensity and brightness of
the image, segmenting objects from each other; Sime and Ferguson, 2003; Detert and Weitbrecht, 2013), and the
other one analyses the textural properties of the whole image, using autocorrelation and semi-variance methods
to define empirical relationship between image texture and the grain sizes of the photographed sediments ( Rubin,
2004; Verdú et al., 2005). Both image processing approaches were very time consuming and required mostly site-
specific manual settings, however, a few transferable and more automated techniques have also been developed
recently (e.g., Graham et al., 2005; Buscombe, 2013). Even though there is a continuous improvement in the
applied image-based bed sediment analysis methods, there are still major limitations the users face with, such as:

• Most of the studies (all the ones listed above) focuses on gravel bed rivers, and only a few exceptions
can be found in the literature where sand is also accounted for (texture-based methods;e.g.:
Buscombe, 2013).
• The adaptation environment was typically non-submerged sediment, instead of underwater
conditions (a few exceptions: Chezar and Rubin, 2004; Warrick et al., 2009).

- The computational demand of the image processing is high (e.g., one to ten minutes per image; Detert and Weitbrecht, 2013).

- The analysis requires operator expertise (higher than in case of any conventional method).

- There is an inherent pixel- and image resolution limit ( Buscombe and Masselink, 2008 Cheng, 2015; Purinton and Bookhagen, 2019). The finer the sediment, the higher resolution of the images should be (higher calculation time), or they must be taken from a closer position (smaller area and sample per image).

Nowadays, with the rising popularity of Artificial Intelligence (AI), several Machine Learning (ML) techniques have been implemented in image recognition as well. The main approaches of segmentation contra textural analysis still remain; however, an AI defines the empirical relationship between the object sizes (Igathinatane et al., 2009; Kim et al., 2020) or texture types (Buscombe and Ritchie, 2018) in the images and their real sizes. In the field of river sedimentology a few examples can already be found, where ML (e.g., Deep Learning; DL) was implemented. For instance, Rozniak et al. (2019) developed an algorithm for gravel-bed rivers, performing textural analysis. With this approach, information is not gained on individual grains (e.g., their individual shape and position), but rather the general grain size distribution (GSD) of the whole images. At certain points of the studied river basins, conventional physical samplings (pebble count) were performed to provide real GSD information. Using this data, the algorithm was trained (with ~1000 images) to estimate GSD for the rest of the study site, based on the images. The method worked for areas where grain diameters were larger than 5 mm, and the sediment was well-sorted. The developed method showed sensitivity to sand coverage, blurs, reduced illuminations (e.g., shadows) and white pixels. Soloy et al. (2020) presented an algorithm which used object detection on gravel- and cobble covered beaches to calculate individual grain sizes and shapes. 46 images were used for the model training, however, the number of images were multiplied with data augmentation (rotating, cropping, blurring the images; see Perez and Wang, 2017) to enhance the learning session and increase the input data. The method was able to reach a limited execution speed of a few seconds per $m^2$ and adequately measured the sizes of gravels. Ren et al. (2020) applied an ensemble bagging-based Machine Learning (ML) algorithm to estimate GSD along the 70 km long Hanford Reach of the Columbia River. Due to its economic importance, a large amount of measurement data has been accumulated for this study site over the years, making it ideal for using ML. By the time of the study, 13,372 scaled images (i.e., their millimetre/pixel ratio was known) were taken both underwater and in the dry zones, covering approx. 1 $m^2$ area each. The distance between the image-sampling points was generally between 50-70 m. An expert defined the GSD (8 sediment classes) of each image by using a special, visual evaluation-classification methodology (Delong and Brusven, 1991; Geist et al., 2000). This dataset was fed to a ML algorithm along with their corresponding bathymetric attributes and hydrodynamic properties, simulated with a 2D hydrodynamic model. Then, it was tested to predict the sediment classes based on the hydrodynamic parameters only. The algorithm performed with a mean accuracy of 53%. Even though this method was not image-based (only indirectly, via the origin of the GSD data), it highlighted the possibilities of an AI for a predictive model, using a high-dimensional dataset. Having such a large data of grain size information can be considered exceptional and takes a huge amount of time to gather, even with the visual classification approach they adapted. Moreover, this was still considered spatially sparse information (point-like measurements, 1 $m^2$ covered area/image dozens of meters away from each other). Buscombe (2020) used a set of 400 scaled images

to train an AI algorithm on image texture properties, using another image-processing method (Barnard et al., 2007)
for validation. The algorithm reached a good result for not only gravel, but sand GSD calculation as well,
outperforming an earlier, but promising, texture-based method (wavelet analysis; Buscombe, 2013). In addition,
the method required fewer calibration parameters than the wavelet image-processing approach. The study also
foresaw the possibility to train an AI which estimates the real sizes of the grains, without knowing the scale of
one pixel (mm/pixel ratio) if the training is done properly. The AI might learn unknown relationships between the
texture and sizes if it is provided with a wide variety (images of several sediment classes) and scale (mm/pixel
ratio)) of dataset (however, it is also prone to learn unwanted biases). Recently, Takechi et al. (2021) further
elaborated on the importance of shadow- detection and removal, using a dataset of 500 pictures for training a
texture-based AI, with the help of an object-detecting image-processing technique (Basegrain; Detert and
Weitbrecht, 2013). The previously presented studies, applying ML and DL techniques, significantly contributed
to the development and improvement of surrogate sampling methods, incorporating the great potential in AI.
However, there are still several shortcomings to these procedures. Firstly, none of the image-based AI studies
used underwater recordings, even though the underwater environment offers completely different challenges.
Secondly, the training images were always scaled, i.e., the sizes of the grains could be easily reconstructed, which
is again complicated to accomplish in a river. Lastly, they were not adapted for continuous (i.e., spatially dense)
measurement, but rather focused on a sparse grid-like approach.
The goal of this manuscript is to further investigate the applicability of image processing as a surrogate method,
and attempt to break through or go around the above mentioned shortcomings of the AI-based approaches. Hence,
we introduce a riverbed material analysing, Deep Learning (DL) algorithm and field measurement methodology,
along with our first set of results. The introduced technique can be used to measure the gravel and sand content of
the submerged riverbed surface. It aims to eventually become a practical tool for exploratory mapping, by
detecting sedimentation features (e.g., deposition zones of fine sediment, colmation zones, bed armour) and
helping decision making for river sedimentation management. Also, the long-term hypothesis of the authors
includes the creation of an image-based measurement methodology, where underwater videos of the riverbed
could serve multiple sediment related purposes simultaneously. Part of which is the current approach for mapping
the riverbed material texture and composition. Others include measuring the surface roughness of the bed (Ermilov
et al., 2020) and detecting bedload movement (Ermilov et al., 2022).
Compared to the studies introduced earlier, the main novelty of our manuscript is that both the training and
analysed videos are recorded underwater, continuously along cross-sections of a large river. Furthermore, the
training is unscaled, i.e., the camera-riverbed distance could vary while recording the videos, without considering
image-scale. Moreover, compared to the relatively low number of training images in most of the above referred
studies, we used a very large dataset (~15000) of sediment images for the texture-based AI, containing mostly
sand, gravel, cobble, and to a smaller extent: bedrock together with some other, non-sediment related objects.

## 2 Methods

### 2.1 Case studies

The results presented in this study are based on riverbed videos taken during three measurement campaigns, in sections of the Danube River, Hungary. The first campaign was at Site A, Ercsi settlement (~ 1606 rkm) where 3 transects were recorded, the second one was at Site B, Gönyű settlement (~ 1791 rkm) with 2 transects, and the third was at Site C, near to Göd settlement (~ 1667 rkm) with 2 transects (Fig. 1). Each transect was recorded separately (one video per transect), therefore our dataset included a total of 8 videos.

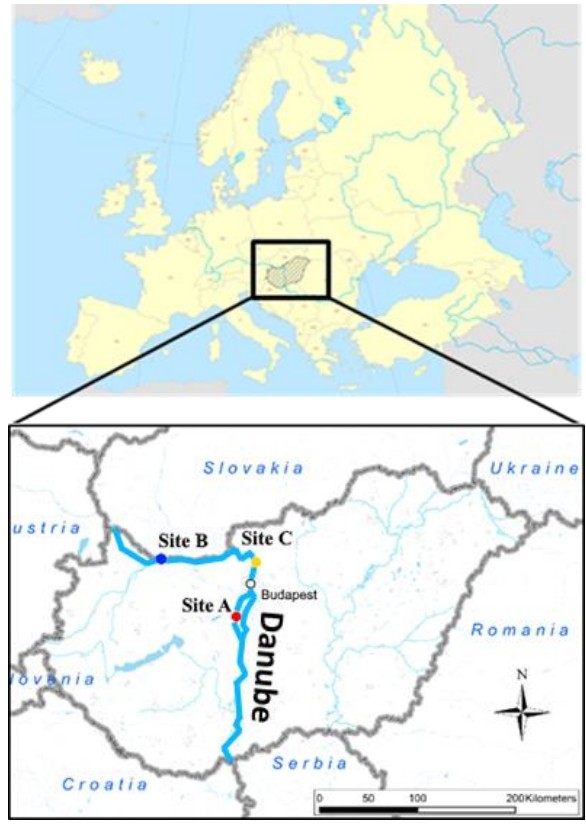

**Figure 1: The location of the riverbed videos, where the underwater recordings took place. All sites were located in Hungary, Central Europe. The surveys were carried out on the Danube River, Hungary's largest river.**

The training of the DL algorithm was done using the video images of Site C and a portion of A (test set; see later in Section 2.3), while Site B and the rest of the images from A served for validation. The measurements were carried out during daytime, at mid-water regime (Q = 1900 m³/s) in case of Site A, and low water regime (Q = 1350 m³/s) at Site B, and Site C (Q = 700 m³/s). This latter site served only for increasing the training image dataset (i.e., conventional samplings were not carried out at the time of recording the videos), thus we do not go into further details with it for the rest of the manuscript, but the main characteristics are listed in Table 1.

|  | Site A | Site B | Site C |
|---|---|---|---|
| $Q_{survey}$ [m³/s] | 1900 | 1350 | 700 |
| $B_{survey}$ [m] | 300 – 450 | | |
| $H_{mean,\ survey}$ [m] | 3.5 - 4.5 | | |
| $S_{survey}$ [cm/km] | 15 | | |
| $SSC_{survey}$ [mg/l] | 25 | 20 | 14 |

| Characteristic riverbed sediment | gravel, sandy gravel | gravel, gravelly sand | gravel, sandy gravel |
|---|---|---|---|
| $Q_{annual,mean}$ [m$^3$/s] | 2000 | 2200 | 1400 |
| $Q_{1\%}$ [m$^3$/s] | 5300 | 5500 | 4700 |

Table 1: Main hydromorphological parameters of the measurement sites. $Q_{survey}$: discharge during survey; $B_{survey}$: river width during survey; $H_{mean,survey}$: mean water depth during the survey; $S_{survey}$: riverbed slope during survey; $SSC_{survey}$: mean suspended sediment concentration during the survey; $Q_{annual,mean}$: annual-mean of the discharge at the site; $Q_{1\%}$: discharge of 1% probability.

As underwater visibility conditions are influenced by the suspended sediment ($SSC_{survey}$ – susp. sed. concentration), the characteristics of this sediment transport is also included in Table 1. The highest water depths were around 6-7 m in all cases. In Site A, measurements included mapping of the riverbed with a camera along three separate transects (Fig. 2a). At Site B, two transects were recorded (Fig. 2b).

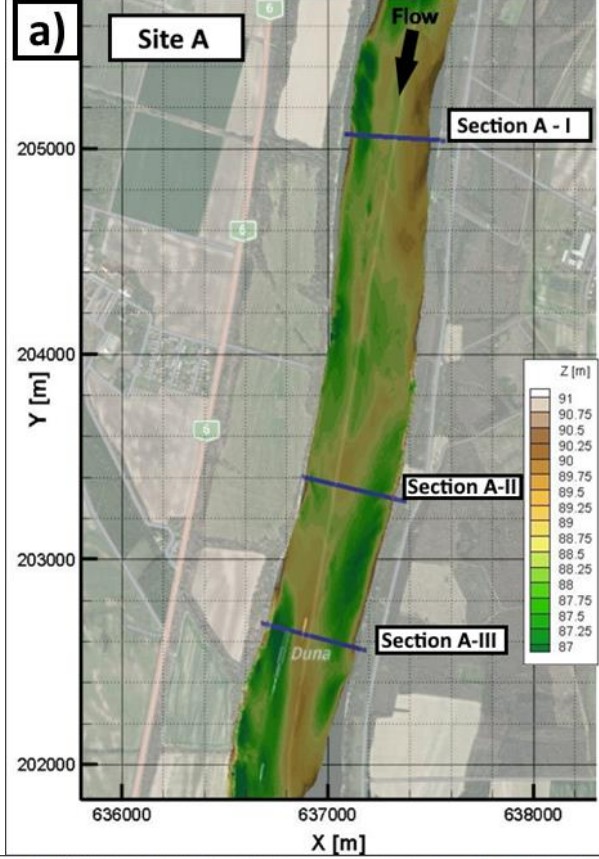

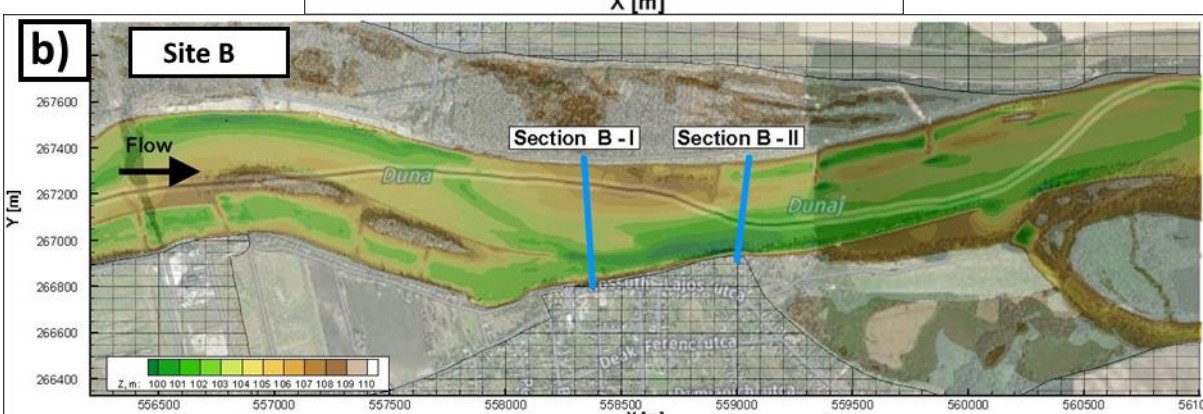

219

220

**Figure 2: Bathymetry of Site A and B. The measurement cross-sections are also marked. The vessel moved along these lines from one bank to the other, while carrying out ADCP measurement and recording riverbed videos. Physical bed material samples were also collected in certain points of these sections.**

224

### 2.2 Field data collection

Fig. 3 presents a sketch of the measurement process with the equipment and a close-up of the underwater instrumentation. During the field measurements, the camera was attached to a streamlined weight (originally used as an isokinetic suspended sediment sampler) and lowered into the water from the vessel by an electric reel. The camera was positioned perpendicularly to the water and the riverbed, in front of the nose of the weight. Next to the camera, two diving lights worked as underwater light sources, focusing into the camera's field of view (FoV). In addition, four laser pointers were also equipped in hand-made isolation cases to provide possible scales for secondary measurements. They were also perpendicular to the bottom, projecting their points onto the underwater

camera field of view. Their purpose was to ensure a visible scale (mm/pixel ratio) in the video footages for validation. During the measurement procedure, a vessel crossed the river slowly through river transects, while the position of the above detailed equipment was constantly adjusted by the reel. Simultaneously, ADCP and RTK GPS measurement were carried out by the same vessel, providing water depth, riverbed geometry, flow velocity, ship velocity and position data. Based on this information and by constantly checking the camera's live footage on deck, the camera was lowered or lifted to keep the bed in camera sight and avoid colliding with it. The sufficient camera – riverbed distance depended on the suspended sediment concentration near the bed and the used illumination. The reel was equipped with a register, with its zero adjusted to the water surface. This register was showing the length of cable already released under the water, effectively the rough distance between the water surface and the camera (i.e., the end of the cable). Of course, due to the drag force this distance was not vertical, but this value could be continuously compared to the water depth measured by the ADCP. Differencing these two values, an approximation for the camera – riverbed distance was given all time. The sufficient difference could be established by monitoring the camera footage while lowering the device towards the bed. This value was then to be maintained with smaller corrections during the survey of the given cross-section, always supported by observing the camera recording, and adjusting to environmental changes. The vessel's speed was also adjusted based on the video and slowed down if the video was blurry or the camera got too far away from the bed (see later in Section 3.3). The measurements required three personnel to i) drive the vessel, ii) handle the reel, adjust the equipment position, and monitor the camera footage, iii) monitor the ADCP data, while communicating with the other personnel (see Fig. 3).

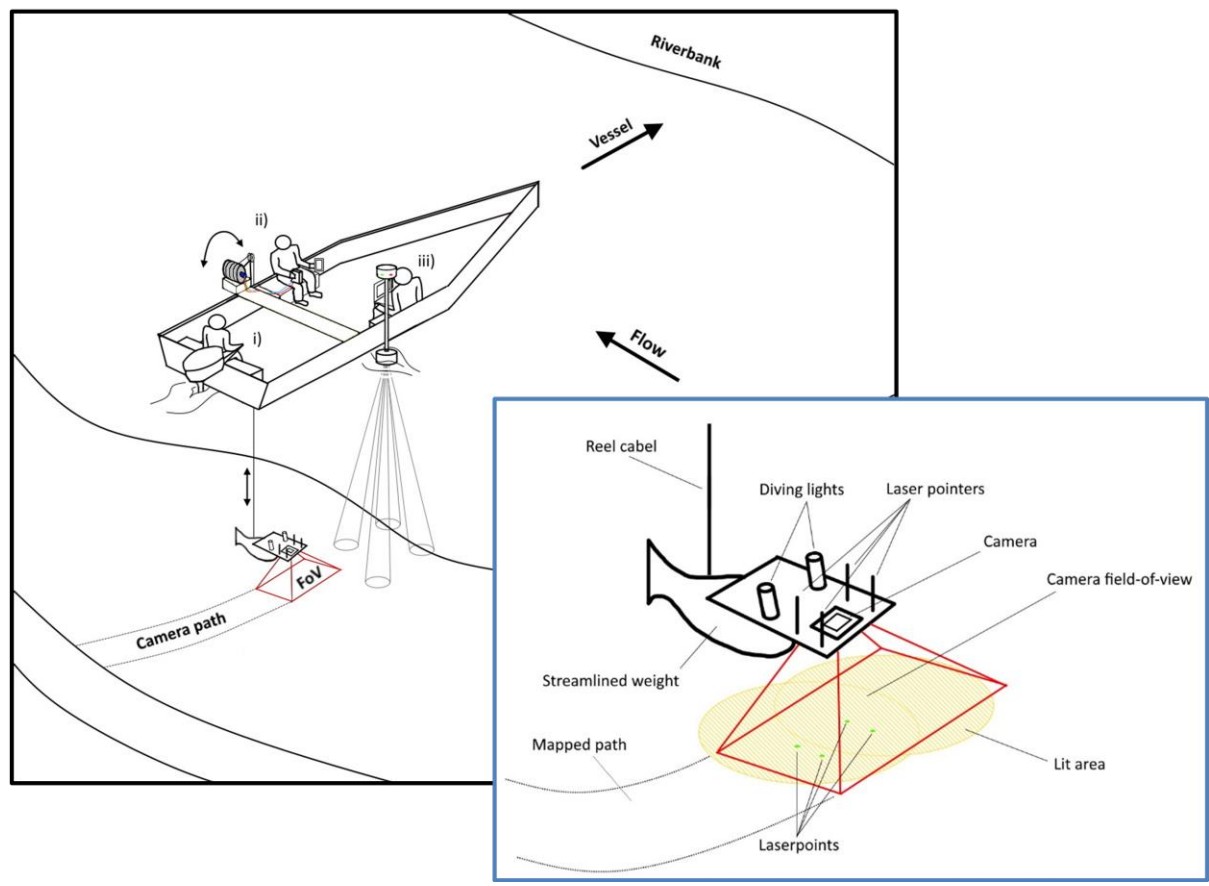

**Figure 3: Left: sketch of the measurement process. The vessel was moving perpendicular to the riverbank along a cross-
section (i). A reel was used to lower a camera close to the riverbed (ii). Simultaneously, the bed topography and water
depth were measured by an ADCP (iii). Right: Close-up sketch of the underwater instrumentation.**
The video recordings were made with a GOPRO Hero 7 and a Hero 4 commercial action cameras. Image
resolutions were set to 2704x2028 (2.7K) with 60 frame per second (fps) and 1920x1080 (1080p) with 48 fps,
respectively. Other parameters were left at their default (see GOPRO 2014; 2018), resulting in slightly different
quality of produced images between the two cameras. We found that a 0.2-0.45 m/s vessel speed with 60 fps
recording frequency was ideal to retrieve satisfactory images in a range of 0.4-1.6 m camera-bed distances. This
meant approximately 15 minutes long measurements per transects. Further attention needed to be paid to the reel
and its cable during the crossing when the equipment was on the upstream side of the boat. If the flow velocities
are relatively high (compared to the total submerged weight of the underwater equipment), the cable can be pressed
against the vessel-body due to the force from the flow itself, causing the reel cable to jump to the side and leave
its guide. This results in the equipment falling to the riverbed and the measurement must be stopped to reinstall
the cable. For illumination, a diving light with 1500 lumen brightness and 75° beam divergence, and one with
1800 lumen and 8° were used. The four lasers for scaling had 450-520 nm (purple and green) wavelength and 1-
5 mW nominal power. Power supply was ensured with batteries for all instruments.
At Site A and Site B, conventional bed material (physical) samplings were also carried out by a grabbing (bucket)
sampler along the analysed transects. At each cross-section 4-5 samples were taken, with one exception where we
had 10. The measured GSDs were used to validate results of the AI algorithm. Separately, a visual evaluation of
the videos was also carried out, where a person divided the transects into subsections based on their dominant
sediment classes, after watching the footages.

**2.3 Image analysis: Artificial Intelligence and the wavelet method**

In this study, we built on the former experiences of the authors, using Benkő et al., 2020 as a proof-of-concept, where the developed algorithm was applied for analysing drone videos of a dry riverbed. The same architecture was used in this manuscript, which is based on the widely used Google's DeeplabV3+ Mobilnet, in which many novel and state-of-the-art solutions are implemented (e.g., Atrous Spatial Pyramid Pooling; Chen et al., 2018). The model was implemented with Pytorch, exploiting its handy API and backward compatibility. The main goal was to build a deep neural network model which can recognise and categorise (via semantic segmentation; Chen et al., 2018) at least three main sediment size classes, i.e., sand, gravel and cobble, in the images, while being quickly deployable. The benefit of the introduced method compared to conventional imagery methods lies in the potential of automation and increased speed. If the annotation and training is carried out thoroughly, analysing further videos can run effortlessly, while the computation time can be scaled down either vertically (using stronger GPUs) or horizontally (increasing the number of GPUs; if parallel analysis of images is desired). In this study a TESLA K80 24GB GDDR5 348bit GPU, an Intel Skylake Intel® Xeon® Gold 6144 Processor (24.75M Cache, 3.50 GHz) CPU with 13GB RAM was used. Also, contrary to other novel image-processing approaches in riverine sediment research (Buscombe, 2013; Detert and Weitbrecht, 2013), the deep convolutional neural network is much less limited by image resolution and mm/pixel ratios, because it does not rely on precise pixel count. This is an important advantage to be exploited here, as we perform non-scaled training and measurements with the AI, i.e., camera-bed distance constantly changed, and size-reference was not used in the images by the AI.

Fig. 4 presents the flowchart of our DL-based image processing methodology. The first step after capturing the videos was to cut them into frames, during which the videos were exploded into sequential images. Our measurement setup proved to be slightly nose-heavy. Due to this, and the drag force combined, the camera tilted forward during the measurements. As a result, the lower parts of the raw images were sometimes too dark, as the camera was looking over the riverbed, and not at the lit part of the bed. In this manuscript, this problem was handled by simply cutting out the lower 25% of the images as this was the region usually containing the dark, unlit areas. Brightening and sharpening filters were applied on the remaining part of the images to improve their quality. Next, the ones with clearest outlines and best visibility were chosen. This selection process was necessary because this way the delineation process (learning the prominent characteristics of each class) can be executed accurately, without the presence of misleading or confusing images, e.g., blurry or dark pictures where the features are hard to recognise. For training purposes, we chose three videos from different sections each being ~15 minutes long with 60 fps and 48 fps, resulting in 129 600 frames. In fact, such a large dataset was not needed due to the strong similarity of the consecutive frames. The number of images to be annotated and augmented were therefore

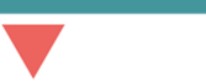

**Data creation**
- Using underwater videos
- Large number of images
- Not fixed elevation from riverbed
- With multiple frame rate
- Selecting images according to variance and visibility

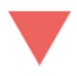

**White Balance Upgrade**
- Balance correction on images with low visibility
- Histogram equalisation with Simple Colour Balance for enhancing edges

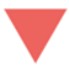

**Data annotation**
- Separating 10 different classes in the images: silt-clay, sand, gravel, cobble, boulder, bedrock, clam-upside, clam-downside, vegetation, unidentified

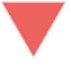

**Data augmentation**
- Increasing data size by: mirroring cropping, rotating, darkening, sharpening, bluring, white- and colour balancing images

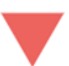

**Training**
- On powerful virtual machines with Tesla K80 GPUs
- 80/20 training/test split
- 14784 images were used. 11827 for training and 2957 for validation

**Visualization and analysis**
- Overall pixel accuracy: 96.35% (mean percent of pixels classified correctly per image during validation)
- Mean IoU: 41.46%
- Overlaying image masks prepared to show the precision of the method
- Comparing DL results on new data with other sampling methods

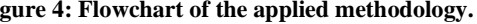

**Figure 4: Flowchart of the applied methodology.**

decreased to ~2000. We also performed a white balance correction on some of the images to improve visibility, making it even easier to later define the sediment class boundaries. We used an additional algorithm to generate more data, with the so-called Simplest Colour Balance method (Limare et al., 2011). It is a simple, but powerful histogram equalisation algorithm which helps to equalise the roughness in pixel distribution.

These steps were followed by the annotation, where we distinguished ten classes: silt-clay, sand, gravel, cobble, boulder (mainly ripraps), bedrock, clam-upside, clam-downside, vegetation, unidentified (e.g., wreckages). Annotation was carried out by a trained personnel, not by the authors, and performed with the help of an open-source software called PixelAnnotationTool (Breheret, 2017), which enables the user to colour mask large parts of an image based on colour change derivatives (i.e., colour masking part of the images which belong to the same class, e.g., purple/red – sand, green – gravel, yellow – cobble, etc.). The masks and outlines were drawn manually, together with the so-called watershed annotation. That is, when a line was drawn, the algorithm checked for similar pixels in the vicinity and automatically annotated them with the same class. The annotation was followed by a data augmentation step where beside mirroring, cropping, rotating the images (to decrease the chance of overfitting), we also convolved them with different filters. These filters added normally distributed noise to the photos to influence the watershed algorithm and applied sharpening, blurring, darkening, and white balance enhancement. Thus, at the data level, we tried to ensure that any changes in water purity, light, and transparency, as well as colour changes, were adequately represented during training. Images were uniformly converted to 960x540 resolution, scaling them down to make them more usable to fit in the GPU's memory. The next step was to convert all the images from RGB (Red-Green-Blue) based colour to grayscale. This is important because colour images have 3-channels, so that they contain a red, a green, and a blue layer, while grayscale images' pixel can only take one value between 0 and 255. With this colour conversion we obtained a threefold increase in computational speed. In total, a dataset of 14,784 human-annotated images was prepared (from the ~2000 images of the 3 training videos). The next step was to separate this dataset into training and validation sets. In this study, 80% of it was used for training the Deep Learning algorithm, while 20% was withhold and reserved for the validation of the training. It was important to mix the images so that the algorithm selects batches in a pseudorandom manner during training, thus preventing the model from being overfitted.

Finally, after several changes in the hyperparameters (i.e., tuning), the evaluation and visualisation of the training
results were performed. Tuning is a general task to do when building Deep Learning Networks, as these
hyperparameters determine the structure of the network and the training process itself. Learning rate, for example,
describes how fast the network refreshes, updates itself during the training. If this parameter is set too high, the
training process finishes quickly, but convergence may not be reached. If it is too low, the process is going to be
slow, but it converges. For this reason, nowadays the learning rate decay technique is used, where one starts out
with a large learning rate, then slowly reduces it. The technique generally improves optimization and
generalization of the Deep Learning Networks (You et al., 2019). In our case, learning rate was initialised to 0.01,
with 30000 iteration steps, and the learning rate was reset after every 5000 iterations with a decay of 0.1. Another
important parameter was the batch size, which sets the number of samples fed to the network before it updates
itself. Theoretical and empirical evidence suggest that learning rate and batch size are highly important for the
generalization ability of a network (He et al., 2019). In our study, a batch size of 16 was used (other general values
in the literature are 32, 64, 128, 256). We used a cross-entropy loss function.

As previously mentioned, the training of the DL algorithm was managed without scaling, without the need for
equipped lasers. However, we intended to use the laser pointers to provide a spatial scale for the recorded videos,
as a secondary validation. As the lasers were not functioning as we originally hoped, we could not use them
constantly during the cross-sectional surveys and could not aim for transactional scaling and validation this way.
Instead, we diverted to validation in the points of the physical samplings as we could use the lasers in a few,
selected points only. We used a textural image-processing method to analyse the video images of these sampling
spots. For this, the already mentioned, transferable wavelet-based signal- and image-processing method was
chosen. The method enables to calculate the image-based grain size distribution of the selected pictures. The grey-
scale intensity is analysed through pixel-rows and -columns of the image and handled as individual signals. Then,
instead of Fourier-transform, the less-constrained wavelet-transform is applied to decompose them. Finally,
calculating the power spectra and the sizes (from pixel to millimetre, using the scale) of the wavelet components
(each wavelet describes an individual grain) produces the grain size distribution for the given image. Beforehand,
this method was proved to be the most efficient, non-DL image-processing method for mixed sediments
(Buscombe, 2013; 2020) and was already tested for underwater circumstances in an earlier study (Ermilov et al.,
375   2020).

**3 Results and discussion**
**3.1 Evaluation of the training**
To evaluate the training process, the 2957 images of the validation set were analysed by the developed Deep
Learning algorithm and the given DL-results were then compared to their human-annotated counterparts. Fig. 5a-
d shows results of original images (from the validation set), their ground truth (annotation by the training
personnel), as well as the DL prediction (result of the model). The overlays of the original and the predicted
images are also shown for better visualization. Calculating the overall pixel accuracy (i.e., the percent of pixels
that were correctly classified during validation) returned a satisfactory result with an average 96% match (over
the 2957 validation images, each having 960x540 resolution, adding up to a total of 1 532 908 800 pixels as
100%). As this parameter in object detection and Deep Learning is not a stand-alone parameter (i.e., it can still be
high even if the model performs poorly), the mean IoU (intersection-over-union or Jaccard index) was also
assessed, indicating the overlap of ground truth area and prediction area, divided by their union (Rahman and
Wang, 2016). This parameter showed a much slighter agreement of 41.46%. Interestingly, there were cases, where
the trained model gave better result than the annotating personnel. While this highlighted the importance of
thorough and precise annotation work, it also showcased that the number of poor annotations was relatively low,
so that the algorithm could still carry out correct learning process and later detections, while not being severely
affected by the mistake of the training personnel. Fig. 5e showcases an example for this: the correct appearance
of cobble (yellow) in the prediction, even though the user (ground truth) did not define it during annotation. As a
matter of fact, these false errors also decrease the IoU evaluation parameter, even though they increase the
performance of the DL algorithm on the long term. Hence, this shows that pure mathematical evaluation may not
describe the model performance entirely. Considering that others also reported similar experience with Deep
Learning (Lu et al., 2018) and the fact that 40% and 50% are generally accepted IoU threshold values (Yang et
al., 2018; Cheng et al., 2018; Padilla et al., 2020), we considered the 41.46% acceptable, while noting that the
annotation and thus the model can further be improved. The general quality of our underwater images may have
also played a role in lowering the IoU result.

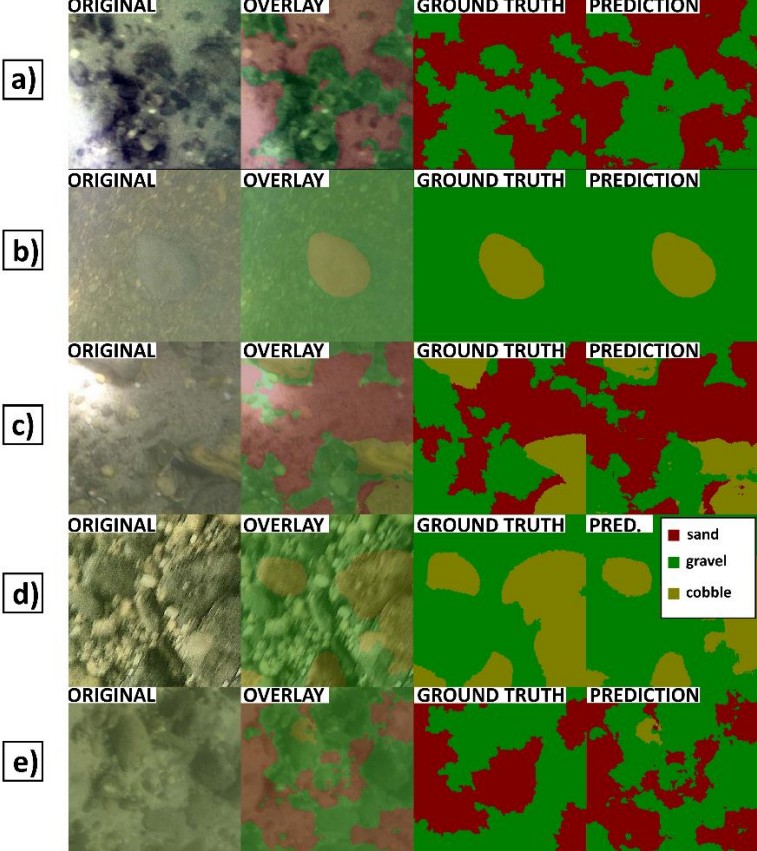

**Figure 5: a-d) Example comparisons of ground truth (drawn by the annotating personnel, 3rd column) and DL**
**predicted (result of analysing the raw image by the previously trained DL model, 4th column) during the validation**
**process. The 1st column shows raw images, while the 2nd column overlays the result of the DL detection on the raw**
**image for better visual context. e) Example of training personnel mistake during the annotation (i.e., lack of**
**cobble/yellow annotation in ground truth) and how the DL performed better by hinting at the presence of the cobble**
**fraction, leading to a false negative result during validation.**

One of these quality issues for the DL algorithm was associated with the illumination. Using a diving light with
small beam divergence proved counterproductive. The high intensity, focused light occasionally caused
overexposed zones (white pixels) in the raw bed image, misleading the DL algorithm and resulting in detection
of incorrect classes there (Fig. 6a). In darker zones, where the suspended sediment concentration was higher and
at the same time, the effect of camera tilting was not completely removed by preprocessing, the focused light
sometimes reflected from the suspended sediment itself and resulted in brighter patches in the images (Fig. 6b).
This also caused false positive detections.

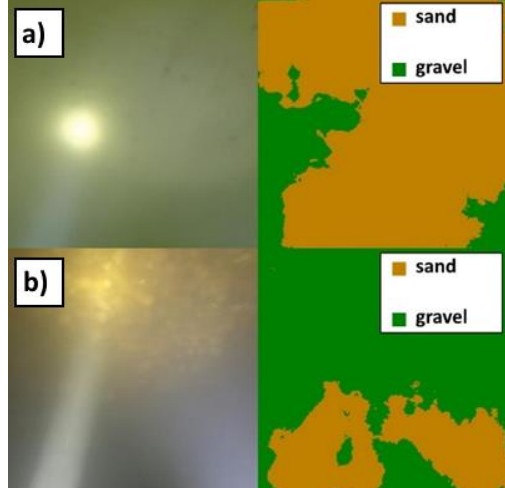

**Figure 6: The effect of strong diving light on the DL algorithm. a) Purely sand covered zone. b) Darker zone with**
**higher SSC. The original images are on the left, while the DL detections can be found on the right.**

**3.2 Comparison of methods**
In each masked image, the occurring percentage of the given class (i.e., the percentage of the pixels belonging to
that class/colour mask, compared to the total number of pixels in the image) was calculated and used as the fraction
percentage in that given sampling point. These sediment classes reconstructed by the DL algorithm were then
compared to three alternative results: i) visual estimation, ii) GSD resulted from conventional grab sampling, iii)
wavelet-based image-processing. In the followings, results from two cross-sections will be highlighted, one from
Site A, the video used for the training, and one from Site B, being new for the DL. An averaging window of 15 m
was applied on each cross-sectional DL result to smoothen and despike the dataset. The interval of physical sample
collection in wider rivers can range anywhere between 20-200 m within a cross-section, depending on the river
width and the homogeneity of riverbed composition. The averaging window size was chosen to be somewhat
lower than our average applied physical sampling intervals in this study, but still in the same order of magnitude.
The scope of the present manuscript did not include further sensitivity analysis of the window size. In the
followings, the reader is led through the comparison process via the example of two transects, and is given the
over-all evaluation of the accuracy of the method.

### 3.2.1 Visual evaluation and physical samples

In Fig. 7a, the path of the vessel can be seen in Section A – II, at Site A. The path was coloured based on the visual evaluation of the riverbed images. The different colours represent the dominant sediment type seen at the given point of the bed. The locations of the physical bed material samplings are also shown (see yellow markers). App. Fig. A1 presents the raw (i.e., before moving-average) results of the DL detection of each analysed image along Section A – II. Currently, our approach is sensitive and large spikes, differences can occur in the DL detection between consecutive, slightly displaced video frames. Due to this, and the fact that there is uncertainty in the coordinates of the underwater photos and their corresponding physical samples, it is not recommended to carry out comparisons by selecting certain image and its DL detection. Instead, we applied a moving average-based smoothing for each raw, cross-sectional DL detection, with a window-size corresponding to 15 m at each site. These moving-averages are later used to compare in the sampling points to the physical sampling and the wavelet method. For illustration purposes, we provided the raw DL detections of all the sampling point images in the Appendix, even though their result may not be representative of their corresponding moving-average values. Fig. 7b shows the cross-sectional visual classification compared to the DL-detected sediment fractions in percentage after applying moving-average (i.e., the smoothed version of App. Fig. A1). The noises are mostly caused by sudden changes in lighting conditions. It happens either from losing visual on the riverbed momentarily due to sudden topography changes or from increased suspended sediment concentration. The DL result shows satisfactory match with the human evaluation. For example, around 100 m from the left bank, between AII-1 and AII-2 sampling points, the DL algorithm peaks with around 70% sand and 30% gravel correctly. Furthermore, on the two side of this peak a steep transition to gravel and decreasing sand occurs, similarly to the visual observation, marked as sandy gravel and gravelly sand. Mixed sediment zones were also correctly identified by the DL algorithm at both riverbanks.

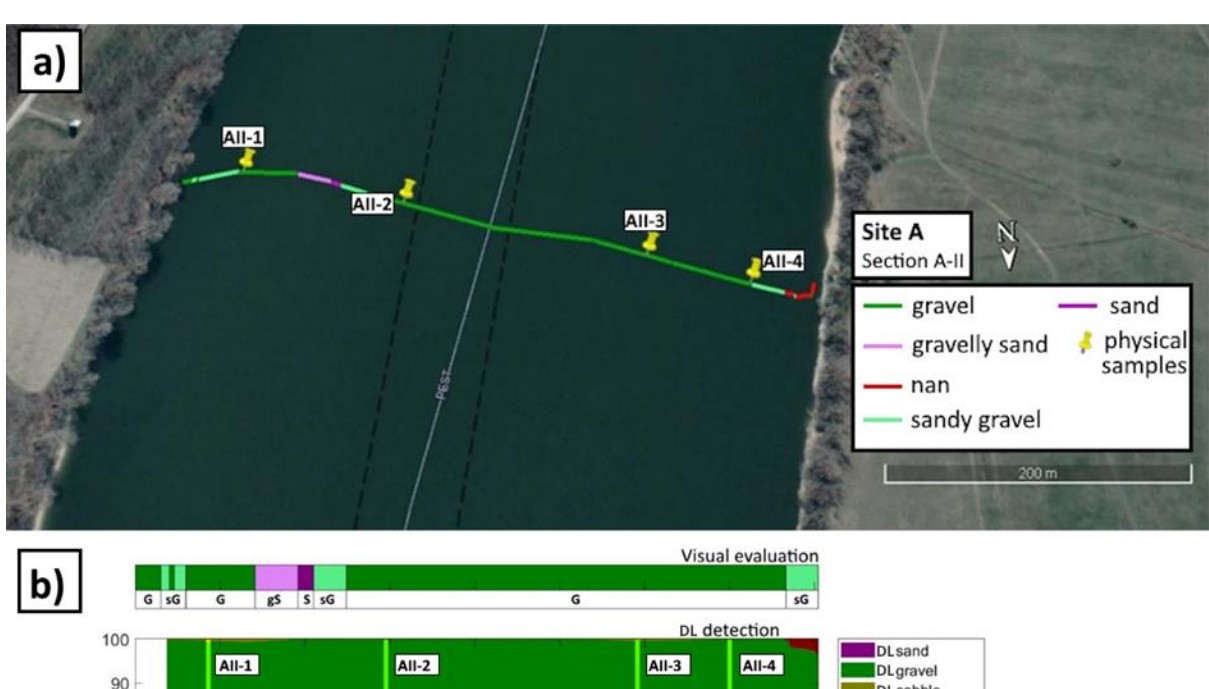

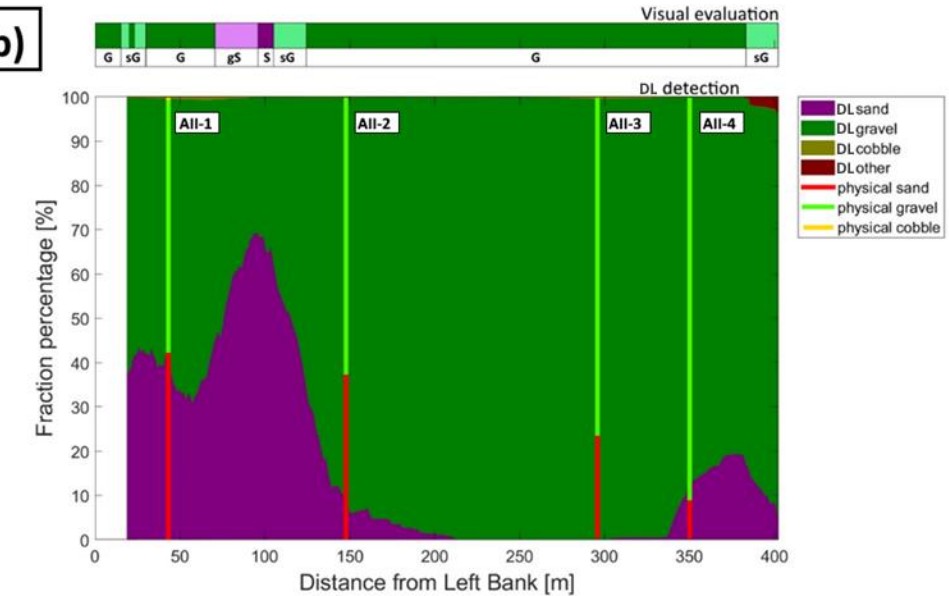


**Figure 7: a) The path of the vessel and camera in Section A – II, Site A. The polyline is coloured based on the sediment**
**features seen during visual evaluation of the video. Yellow markers are the locations of physical bed material samplings.**
**(Map created with Google Earth Pro). b) The visual evaluation of the dominant sediment features in the video (top)**
**compared to sediment fraction percentage, recognised by the DL algorithm (bottom). DL result after applying moving-**
**averaging. The visual evaluation included four classes: gravel – G, sandy gravel – sG, gravelly sand – gS, sand – S,).**
**The fractions of the physical samples are shown as verticals.**

At site B (Fig. 8a) the river morphology is more complex compared to Site A as a groyne field is located along
the left bank (see Fig. 2b). As such, the low flow regions between the groynes yield the deposition of fine
sediments, and much coarser bed composition in the narrowed main stream. As it can be seen, the DL algorithm
managed to successfully distinguish these zones: the extension of fine sediments in the deposition zone at the left
bank were adequately estimated and showed a good match with the visual evaluation for the whole cross-section
(see Fig. 8b).

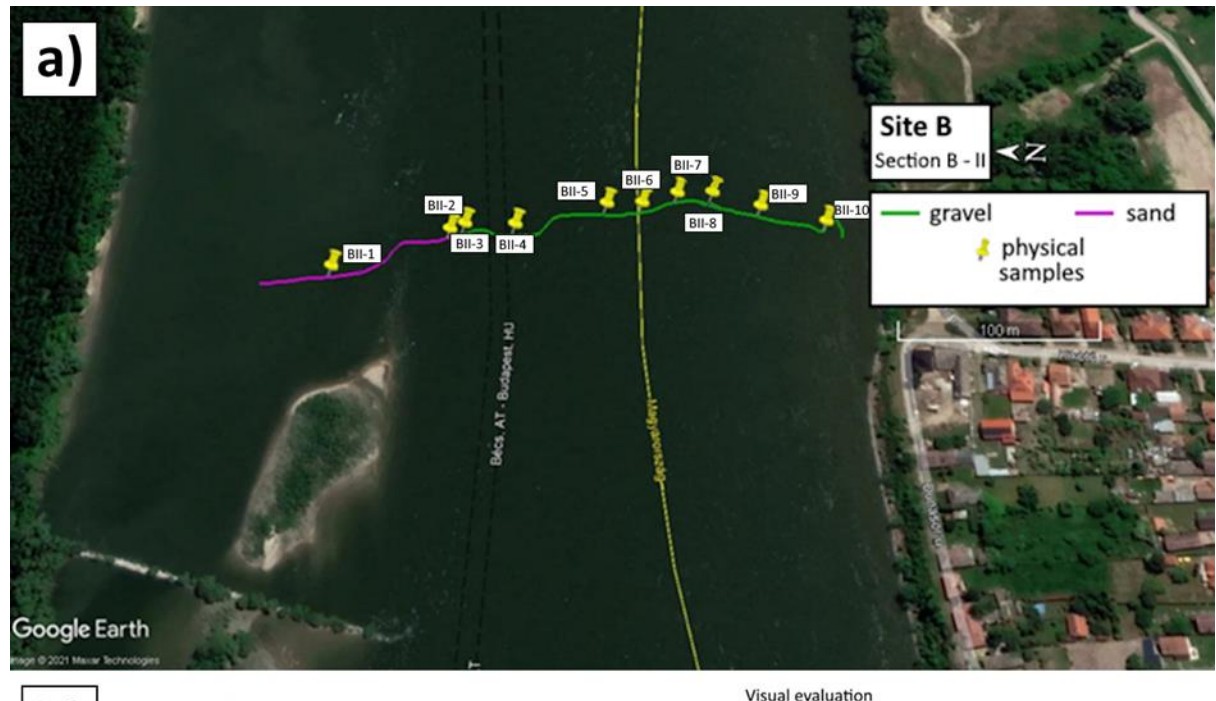

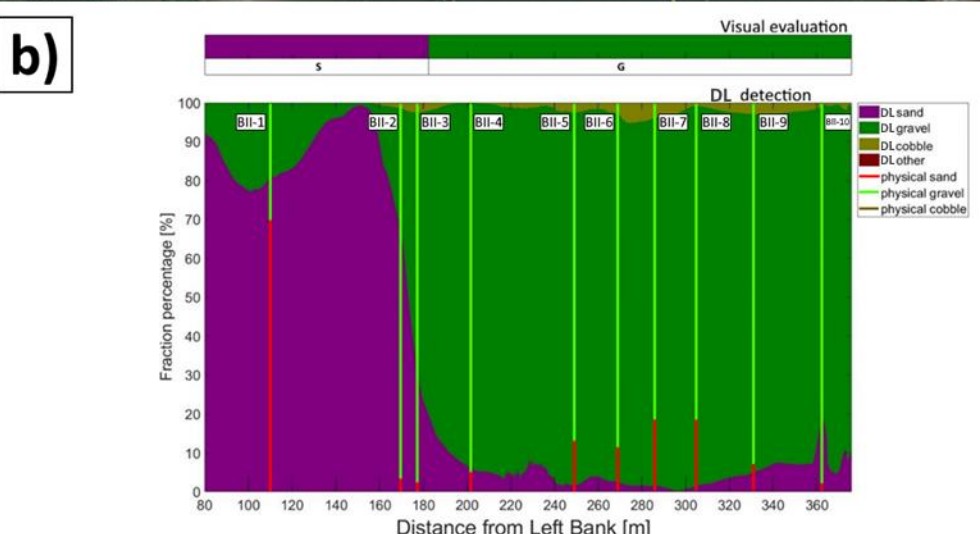

**Figure 8: a) The path of the vessel and camera in Section–B - II, Site B. The polyline is coloured based on the sediment seen during visual evaluation of the video. Yellow markers are the locations of physical bed material samplings. (Map created with Google Earth Pro). b) Sediment fraction percentages in Section–B - II, recognised by the AI. The visual evaluation included two classes: gravel – G, sand – S). The fractions of the physical samples are shown as verticals.**

Results of the other measurements can be found in the Appendix. App. Fig. C2, D2 and E2 show that the trend of riverbed composition from the visual evaluation is well-captured by the DL algorithm in the other cross-sections of the study as well.

Next, the physically measured and DL-detected relative proportion of sand, gravel and cobble fractions were compared in each of the 27 sampling points. Firstly, however, outliers had to be identified. In our case, this meant the separation of sampling points where the differences between the results of the two method were independent from the efficiency and performance of the DL algorithm. This selection was carried out after analysing the grainsize distribution curves of the weight-sieved physical samples (App. Fig. F1) and the riverbed images at the sampling points (App. Fig. A3, B1, C4, D4, E4). Based on our findings, the outliers have been identified and

separated into Outlier Type A, and Outlier Type B categories. First category included the sampling points where the GSD curves showcased bimodal (gap graded) distributions. This type of riverbed sediment distribution is a typical sign of riverbed armouring (Rákóczi, 1987; Marion & Fraccarollo, 1997), where a coarse surface layer protects the underlying finer subsurface substrate (see e.g., Wilcock, 2005). While the camera only sees the upper layer, the bucket sampler can penetrate the surface and gather sample from the subsurface as well. As a result, the two methods cannot be compared solely on the surface distribution. In App. Fig. A2, supportive images of bed armouring are provided, taken during our surveys in the Upper section of the Hungarian Danube. Out of the 27 sampling points, 11 was categorised as Outlier Type A. The category of Outlier Type B consisted of points from the opposite case: where the riverbed image contained fine sediment, but the physical samples did not. In these cases, a relatively thin layer of fine sediment covered the underlying gravel particles. 2 sampling points were categorised as Outlier Type B, both of which were near to the borderline between a deposition zone behind a groyne, and the gravel bedded main channel. In these cases, the bucket sampler probably either stirred up the deposited fine sediment and washed it down during its lifting or was dragged through purely gravel bedded patch during sampling, as the surface composition was rapidly changing on this before-mentioned borderline. It is also worth noting that the physical samples are analysed by weighing the different sediment size classes, resulting in weight distribution. On the other hand, the imagery methods provide surface distributions, hence having a thin layer of fine sediments on the top can strongly bias the resulted composition (Bunte and Abt, 2001; Sime and Ferguson, 2003; Rubin et al., 2007).

| | Comparable data | Outlier Type A | Outlier Type B | $\sum$ |
|---|---|---|---|---|
| **No. sampling points** | 14 | 11 | 2 | 27 |

**Table 2: After evaluating the results of the sieving analyses and riverbed surface images, out of the 27 sampling points, 14 were defined as comparable between the applied sampling methods. 11 points were categorised as Outlier Type A, because their GSD curves were bimodal. 2 points were defined as Outlier Type B, since their images showed the presence of fine sediment, while the sieve analyses did not.**

Overall, the DL-based classification agreed well within the comparable sampling points, with an average error of 4.5% (Fig. 9). It can be seen that even though in outlier points AII-1 and AI-3 the DL algorithm coincidentally gave good match with the sieving analysis, in the rest of the outlier points the DL- and physical-based results systematically differ from each other, supporting our outlier selection methodology.

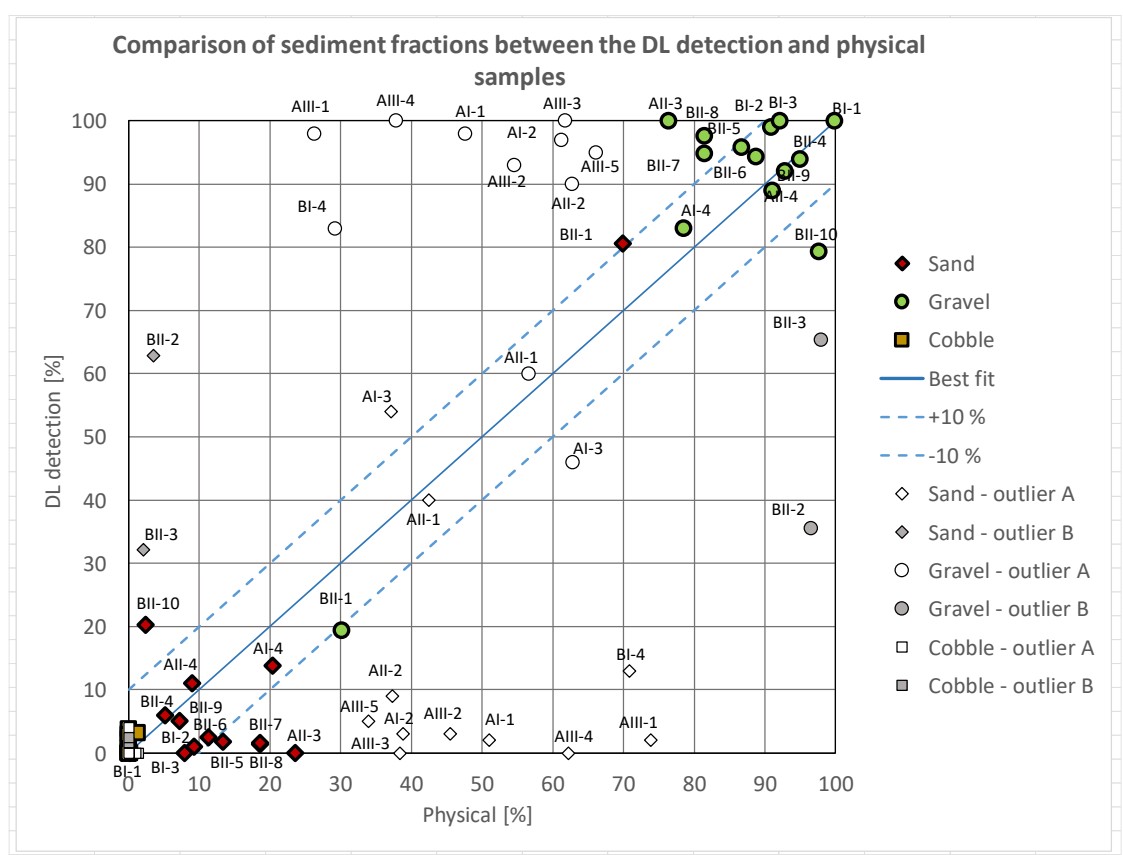

**Figure 9:** *Comparison of* relative *sediment fractions between the DL detection and physical samples*. **The three main sediment types (sand-gravel-cobble) are marked with different colour and symbols. The name of the sampling points where the given relative proportion was measured/detected is also written for gravel and sand (cobble was negligible). The proportions of outlier sampling points are marked with white/grey, while the symbol represents the sediment type respectively. The comparable points have their proportions with green (gravel) and red (sand) symbols.**

### 3.2.1 Wavelet analysis

As for the wavelet analysis-based imagery technique, an overall slight overestimation of the coarse particles can be observed, and the sand classes are, in fact, not reconstructed correctly. This finding agrees well with the field experiences of Ermilov et al. (2020), where the authors indicated the strong sensitivity of the wavelet technique on the image resolution, and showed that to detect a grain, the diameter must be at least three times larger than the pixel. In the following, the wavelet-detected relative sediment proportions are compared to the earlier, corresponding DL-, and physical based ones via bar plots (Fig. 10, 11). For example, the camera was closer to the riverbed at sampling points AII-1 and AII-4, resulting in a better mm/pixel ratio, hence the wavelet algorithm was able to detect coarse sand, but finer sand was neglected yielding the lower sand percentages (Fig. 10). In the other sampling points, where sand was below its resolution, the wavelet method systematically measured the presence of cobbles instead (Fig. 10), even though the other two methods did not. This trend generally described the performance of the wavelet method during our study. For visual purposes, an example of the difference in the capabilities of the two method is given in Fig. 12. While both detected the presence of two major sediment categories, the wavelet translated the information as gravel and cobble mixture, meanwhile the DL algorithm recognised the sand coverage and gravel particles.

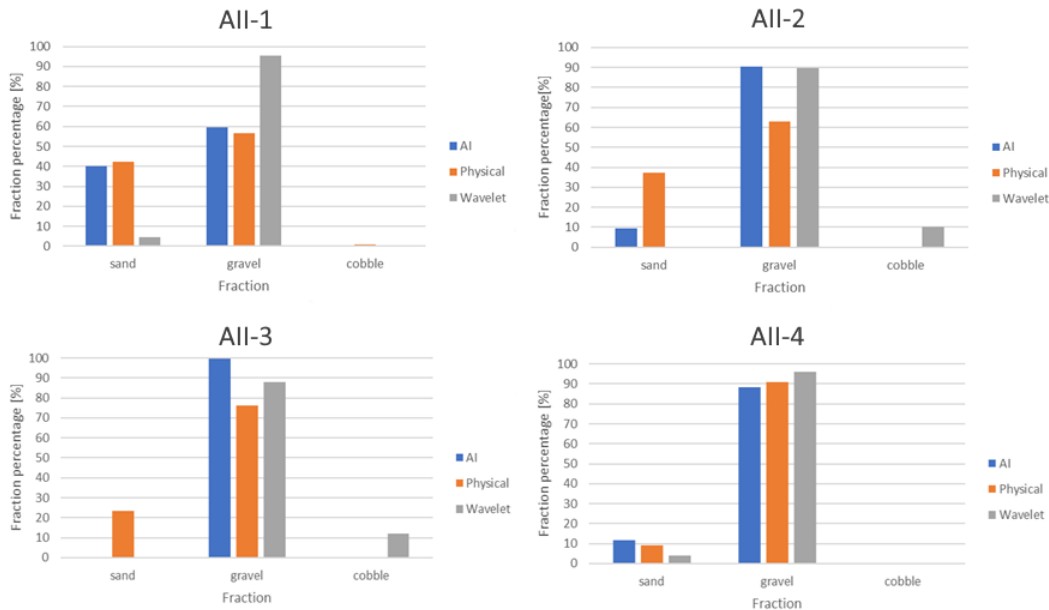

**Figure 10: Comparison of relative sediment fraction proportions [%] at the sampling locations from the moving-**
**averaged DL detection, conventional sieving and the wavelet-based image processing method. Section A – II.**

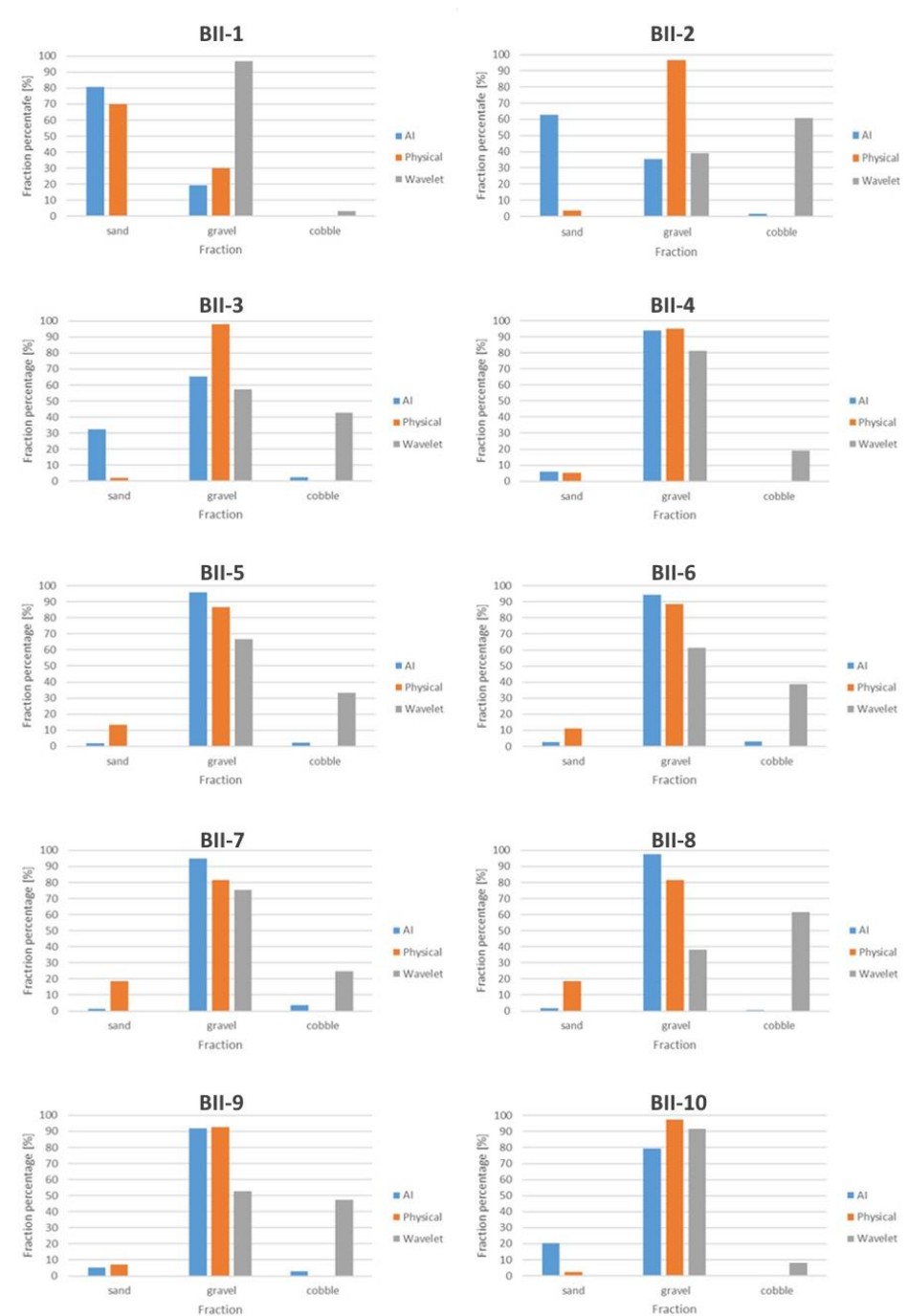

**Figure 11: Comparison of relative sediment fraction proportions [%] at the sampling locations from the moving-**
**averaged DL detection, conventional sieving and the wavelet-based image processing method. Section–B - II.**

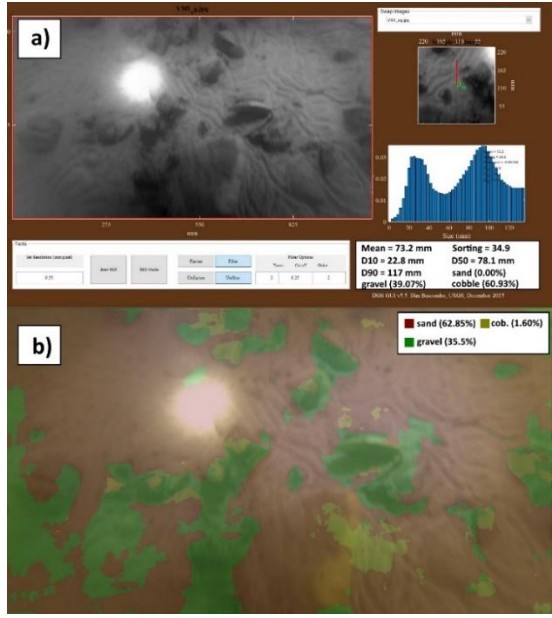


*Figure 12: a) Wavelet analysis result of the underwater image in BII-2. b) DL detection result of the same image.*


Overall, the comparison between the two image-based method showed greater discrepancies (Fig. 13), due to the
limitations of the wavelet approach, discussed earlier. The same sampling points were labelled as outliers as
earlier. As it can be seen, the wavelet significantly differed in the points where the physical samples and DL-
detections matched (green data points), due to its excessive, false cobble detections. However, it showed good
agreement with the DL in most of the outlier points, supporting that the surface in those points was indeed
composed of solely gravel, and the finer fractions of the physical samples must have come from the subsurface.
Hence, our outlier selection process was well based.

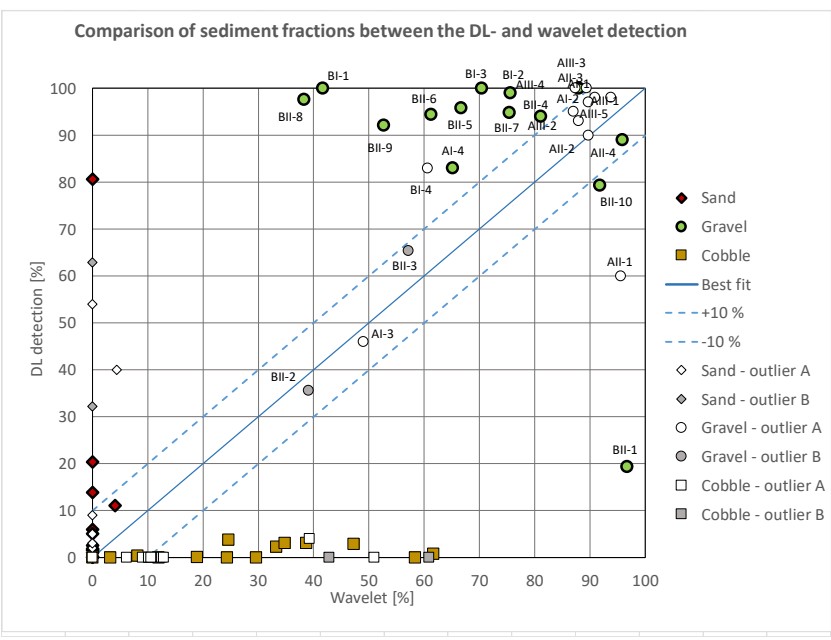


**Figure 13: Comparison of sediment fractions between the DL detection and the wavelet approach, for the selected**
**sampling points. The three main sediment types (sand-gravel-cobble) are marked with different colour (red-green-**
**yellow) and symbols (diamond-circle-square) respectively. The name of the sampling points where the given relative**
**proportion was measured/detected is also written for gravel. The proportions of outlier sampling points are marked**
**with white/grey, while the symbol represents the sediment type respectively. The comparable points have their**
**proportions with green (gravel), red (sand) symbols.**

Based on the results presented in this manuscript, it could be established that the DL algorithm managed to recognise the main features of the riverbed material composition from underwater videos with satisfactory accuracy in the comparable sampling points (compared to the sieving analysis of physical samples) and along cross-sections (based on the visual evaluation). The method showed good potential for mapping heterogenous riverbeds along river cross-sections. Furthermore, the wavelet proved to be a limited comparison tool with the introduced field measurement methodology, as it did not provide it with the sufficient resolution most of the time.

### 3.3 Implementation challenges

The power supply for the entire imaging infrastructure, i.e., for the camera, the diving lights and lasers, was ensured by batteries. However, due to the low temperature at the river bottom, the battery level decreased extremely fast, compared to normal circumstances. Providing the power supply directly from the motorboat engines can overcome this issue. To keep the camera in the adequate height also caused difficulties, since getting too close to the bed can harm the devices, lifting too high, on the other hand, will result in poor image quality. The measured instantaneous ADCP flow depth data was used therefore to keep the bed in camera sight, while maintaining proper boat velocity to avoid blurry images. Choosing a higher recording frequency, however, can be beneficial and alter this limitation, when provided. Lower velocities could not be maintained as the river would have moved the vessel out of the section. An alternative solution can be to move on longitudinal (streamline) paths instead of transects. This would allow for lower vessel speed. This would increase the time of the measurement, which still could be profitable if the images are of higher quality. However, the conventional way for river bathymetry surveys is to move on transversal, cross-sectional paths, due to the river bathymetry having a lower spatial variation along streamlines, compared to the changes that occur in the transversal direction (Benjankar et al., 2015; Kinsman, 2015). As such, it may require carrying out a relatively dense set of longitudinal paths to gain proper information, further increasing the time demand. Thus, for this alternative, higher attention needs to be paid towards choosing these paths and the interpolation method. Another challenge can be the influence of drag force on the measurement setup. In our case, even though the main body itself was a streamlined weight, equipping the other tools on it turned the setup geometry irregular. Additionally, we found that our setup was a bit nose-heavy. On the long term however, this effect could be reduced by building a streamlined container (e.g., 3D-printed body, or a body similar to unmanned underwater vehicles') with slots in it for each device, and also by improving the weight distribution. Furthermore, we hypothesize that by using lasers (as originally planned in this study) during the measurements, the known structure (i.e., the position and distances) of the laser points when the setup is perpendicular to the bed, can help to orthorectify the images. This will decrease the effect of occasional tilting when one wishes to carry out size analysis on the images. In our case, we presented how the wavelet method had inherently bigger issues (i.e., image resolution limit), which could not be caused by the camera tilting since those would be in a significantly lower magnitude of error.

As for the training of the DL algorithm with the underwater images, the illumination is indeed a more crucial aspect, compared to normal imagery methods. In many cases only the centre areas of the images were clearly visible, whereas the remaining parts were rather dark and shady. Determining the boundaries between distinct sediment classes for these images was challenging even for experienced eyes. This quality issue certainly

generated some incorrect annotations. To overcome this issue, manually varying the white balance thus enhancing
the visibility of the sediment could improve the training to some extent. It is worth noting that when Deep Learning
methods are to be used, most of the problems arise from the data side (Yu et al., 2007), whereas issues related to
the applied algorithms and hardware are rare. This is because data is more important from an accuracy perspective
than the actual technical infrastructure (Chen et al., 2020). The time demand of image annotation (data
preparation) is relatively high, i.e., a trained person could analyse roughly 10 images per hour. On the other hand,
as introduced earlier, a great advantage of using DL is the capability of improving the quality of training itself,
often yielding better agreement with reality, compared to the manual annotation. Similar results have been
reported by Lu et al., (2018). This at the same time proves that with the introduced approach, there is no need for
very precise manual training, thus a fast and effective training process can eventually be achieved.
The validation of the Deep Learning algorithm is far from straightforward. In this study, four approaches were
adapted, a mathematical approach, and comparison with three other measurement methods, respectively. The
mathematical approach was based on calculating pixel accuracy and the Intersection-over-union parameter, as it
is usually done in case of Deep Learning methods to describe their efficiency (e.g., Rahman and Wang, 2016).
However, the DL model in some cases overperformed, and provided more accurate results for the sediment
composition than the human annotator did. This meant the calculated difference between the annotated validation
images and their responding DL-generated result was not solely originated from underperformance of the DL-
model, but from human error as well Consequently, using only the mathematical evaluation in this study could
not describe adequately the model performance. Hence, the results were compared to those of three other methods:
i) visual evaluation of the image series, ii) a wavelet-based image-processing method (using the method of
Buscombe, 2013) and iii) riverbed composition data from physical samples. Considering the features of the
applied methods, the first one, i.e., the visual observation, is expected to be the most suitable for the model
validation. Indeed, when assessing the bed surface composition by eye, the same patterns are sought, i.e., both
methods focus on the uppermost sediment layer. On the other hand, the physical sampling procedure inherently
represents subsurface sediment layers, leading to different grain size distributions in many cases. For instance, as
shown above, if bed armour develops in the riverbed and the sampler breaks-up this layer, the resulted sample can
contain the finer particles from the subsurface layer. On the contrary, in zones where a fine sediment layer is
deposited on coarse grains, i.e., a sand layer on the top of a gravel bed, the physical samples represent the coarse
material too, moreover, considering that the sieving provides weight distribution this sort of bias will even enhance
the proportion of the coarse particles. Attempts were made to involve a third, wavelet-based method for model
validation. However, this method failed when finer particles, i.e., sand, characterized the bed. This is an inherent
limitation of these type of methods, as discussed earlier, i.e., the pixel size is simply not fine enough to reconstruct
the small grain diameters in the range below fine gravel. Lastly, the most comparable sample points were selected
to quantify the performance of the DL. Holding the sieved physical samples as ground truth, the DL algorithm
showed promising results. The average error (difference) between DL-detected and physically measured relative
sediment fraction portion percentages was 4.5%. Furthermore, the DL algorithm successfully detected the trend
of changing bed composition along complete river cross-sections.
As it is known, the ML and DL models can learn unknown relationships in datasets, but unwanted biases as well.
With our current dataset, in our opinion, these biases would be the darker tones of visible grain texture and the
lack of larger grain sizes. This way our model in its current state is only applicable effectively in the chosen study
site, until the dataset is not expanded with additional images from other rivers or regions. However, the purpose
of the manuscript was to introduce the methodology itself and its potential in general and not to create a universal
algorithm.
**3.4 Novelty and future work**
The introduced image-based Deep Learning algorithm offers novel features in the field of sedimentation
engineering. First, to the authors' knowledge, underwater images of the bed of a large river have not yet been
analysed by AI. Second, the herein introduced method enables extensive (and still relatively quick) mapping of
the riverbed, in contrast to most of the earlier approaches, where only several points or shorter sections were
assessed with imagery methods. Third, the method is much faster compared to conventional samplings or non-
DL-based image-processing techniques. The field survey of a 400 m long transect took ~15 minutes, while the
DL analysis took 4 minutes (approx. 7 image/s). The speed range of 0.2-0.45 m/s of the measurement vessel and
the 15 minutes per transect complies with the operating protocol of general ADCP surveys on rivers (e.g., RD
Instruments, 1999; Simpson, 2002; Mueller and Wagner, 2013). Hence, the developed image-based measurement
can be carried out together with the conventional boat-mounted ADCP measurements, further highlighting its time
efficiency. Indeed, the method is a great alternative approach for assessing riverbed material on-the-go, in
underwater circumstances. As an extensive and quick mapping tool, it can support other types of bed material
samplings in choosing the sampling locations and their optimal number. Furthermore, it can be used for quickly
detecting areas of sedimentation and their extent, as it was shown in Section 3.2. (e.g., Fig. 12b). This way, it may
support decisions regarding the maintenance of the channel or the bank-infiltrated drinking water production
(detecting colmation and colmated zones). Fourth, a novel approach was used for the imaging and model training.
As the camera-bed distance were constantly changing, the mm/pixel ratio also varied. Hence, no scale was defined
for the algorithm beforehand. Earlier Deep Learning methods for sediment analysis all applied fixed camera
heights and/or provided scaling for the AI. It should be noted that these were airborne measurements, mapping
the dry zone of the rivers. In an underwater manner, it is extremely challenging to keep a fixed, constant camera
height due to the spatially varying riverbed elevations. Hence, it is of major importance that this manuscript
introduces a methodology and a Deep Learning algorithm which neglect the need for scaling. This way, the
method is faster and easier to build, but also simpler to use. Of course, as a trade-off, the method, as of now,
cannot reconstruct detailed grainsize distributions. Indeed, the purpose was rather to provide a uniquely fast bed
material mapping tool, additionally with a much denser spatial resolution than the conventional methods, saving
up significant resources.

Originally, beside the three main sediment grain classes introduced in the manuscript (sand, gravel, cobble), others
were also defined during annotation (e.g., bedrock, clams), but due to class imbalance (i.e., dominance of the three
sediment classes), these were not adapted successfully. There is a good potential in improving the method through
transfer learning (see Zamir et al., 2018) using broader dataset, involving other sediment types. Another possible
way to counter imbalance is the use of so-called weighted cross entropy (see Lu et al., 2019) on the current dataset,
which will also be investigated in our case.

Since the introduced method offers a quick way to provide extensive, spatially dense bed material information of
its composition, it may be used to boost the training dataset of predictive, ensemble bagging-based Machine
Learning techniques (e.g., Ren et al., 2020) and improve their accuracy. Furthermore, the method can support the
implementation of other imagery techniques. For instance, using one of the training videos of this study the authors
managed to reconstruct the grain-scale 3D model of a riverbed section with the Structure-from-Motion technique
(Ermilov et al., 2020), enabling the quantitative estimation of surface roughness. Underwater field cameras can
also be used for monitoring and estimating bedload transport rate (Ermilov et al., 2022) by adapting LS-PIV and
the Statistical Background Model approach. This latter videography technique may also be used with moving
cameras (e.g., Hayman and Ekhlund, 2003), which enables its adaptation into our method by e.g., detecting
bedload movement in the cross-section.

The statistical representativity of the introduced method, as a surface sampling technique, needs to be also
addressed in future work. Following and building upon the experience of conventional, surface sampling
procedures (e.g., grid sampling; Diplas, 1988) may prove to be beneficial, where they provided the exact number
of gravel particles needed to be included (Wolman, 1954) to satisfy the representativity criteria. Then, using edge-
and blob-detection would enable to calculate and compare the number of gravel particles in the images to this
value. Furthermore, we intend to apply 2 cameras, with overlapping FOVs for increasing the covered area (and
the representativity) during surveys. Besides, it would also improve the accuracy of the Structure-from-Motion
technique mentioned earlier.
**4 Conclusion**
A novel, artificial intelligence-based riverbed sediment analysis method has been introduced in this manuscript,
which uses underwater images to reconstruct the spatial variation of the characteristic sediment classes. The
method was trained and validated with a reasonably high number (~15000) of images, collected in a large river,
in the Hungarian section of the Danube. The main novelties of the developed Deep Learning based procedure are
the followings: i) underwater images are used, ii) the method enables mapping of the riverbed along the
measurement vessel's route with very dense spatial allocation, iii) cost-efficient, iv) works without scaling, i.e.,
the distance between the camera and the riverbed can vary. Consequently, in contrast with conventional pointwise
bed sediment analysis methods, this technique is robust and capable of providing continuous sediment
composition data covering whole river reaches, eventually providing the possibility to set up 2D bed material
maps. In this way, river reach scale hydromorphological assessments can be supported, where the composition of
bed surface is of interest, e.g., when performing habitat studies, parameterising 2D and 3D computational
hydrodynamic and morphodynamic models, or assessing the impact of restoration measures.
**Financial support.** The first author acknowledges the support of the ÚNKP-21-3 New National Excellence
Programme of the Ministry for Innovation and Technology, and the National Research, Development and
Innovation Fund, Hungary.
**Code availability.** The code written and used in this manuscript is available here: https://bmeedu-
my.sharepoint.com/:f:/g/personal/ermilov_alexander_emk_bme_hu/EjI2neM4AOZGsBkYgKReViEBBzRFRFo
YyLlmo6SzTB_qDQ?e=AqpqHI
**Data availability.** The dataset and results can be accessed using the following link: https://bmeedu-
my.sharepoint.com/:f:/g/personal/ermilov_alexander_emk_bme_hu/EhoGx64sP1tFnj8Z1OdMZAsBZWd5gDY
zPyodSUDdWFjeiw?e=hKIXjq
**Author contributions.** GB developed the code and carried out the training process. AAE carried out the
fieldwork, evaluated the results, did the laboratory analysis, and collaborated with GB in improving the images.
SB oversaw and directed the project, while managing the financial- and equipment background.
**Competing interest.** The contact author has declared that none of the authors has any competing interest.
**Acknowledgements.** The authors would like to thank our students Dávid Koós, Gergely Tikász, Schrott Márton
and our technicians István Galgóczy, István Pozsgai, Károly Tóth and András Rehák for fieldwork support.

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

**Appendix**
**Appendix A Site A - Section A – II**

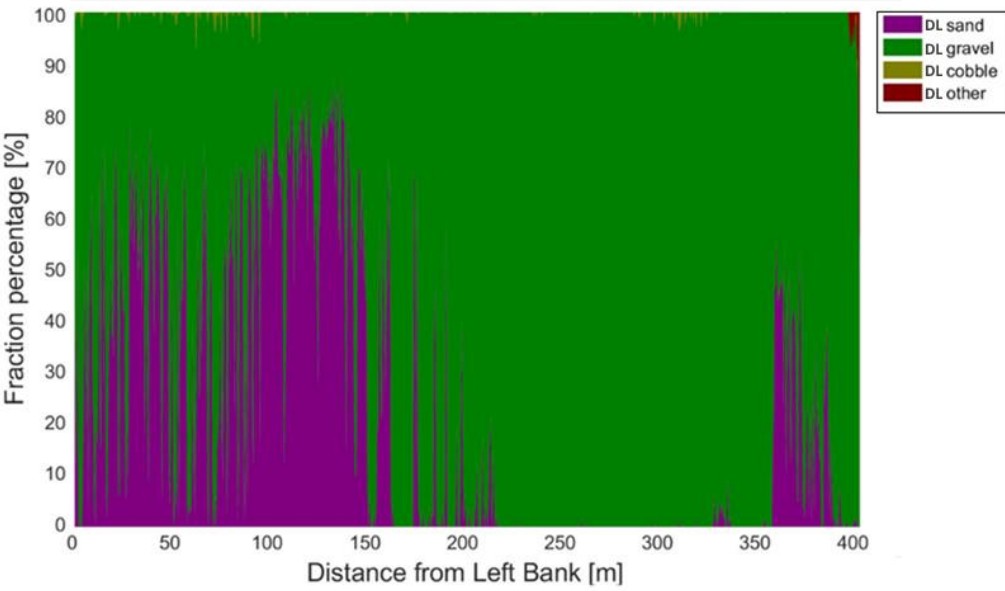

**Figure A1: The sediment fraction percentage results of every image, analysed by the DL algorithm along Section A -**
**II. While the trends are apparent, the sensitivity of the method at its current state can be observed. DL result before**
**applying moving-averaging.**

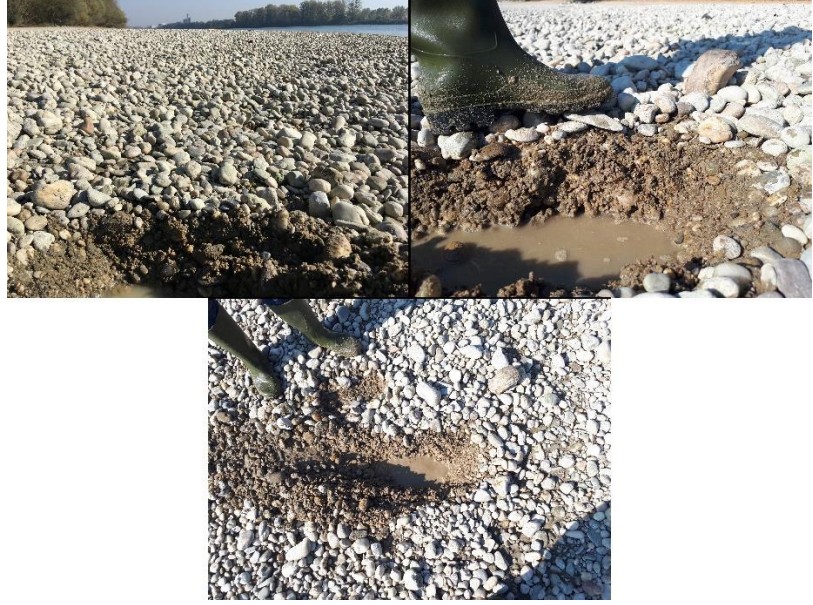

**Figure A2: Images of bed armouring, taken during our surveys in the Upper section of the Hungarian Danube. We**
**broke the surface armour to showcase the presence of the underlying finer fractions.**

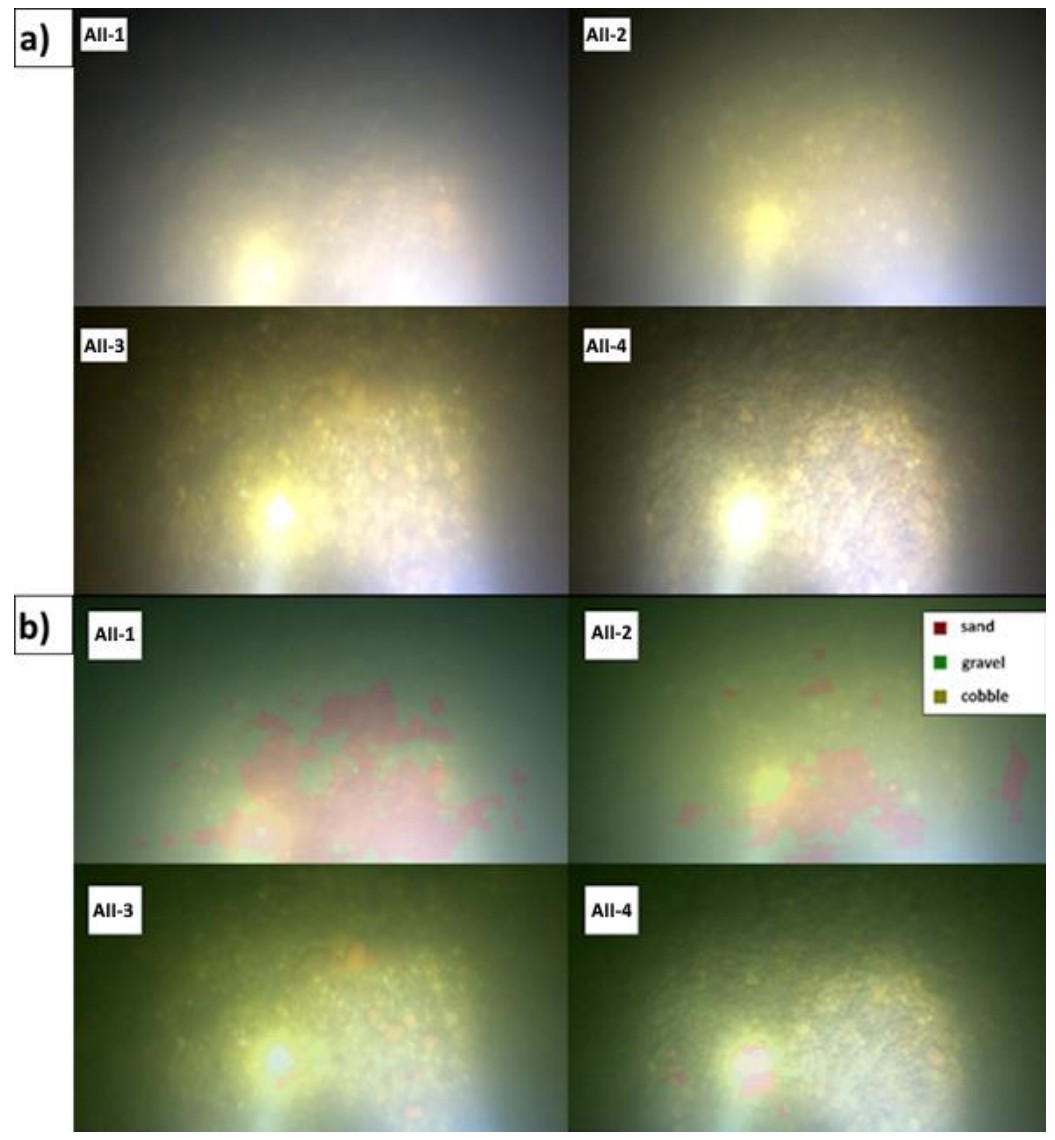

**Figure A3: a) Riverbed video images at the sampling points in Section A - II. b) Riverbed video images overlapped with**
**their raw, DL detection result, at the sampling points in Section A - II.**
**Appendix B Site B - Section B – II**

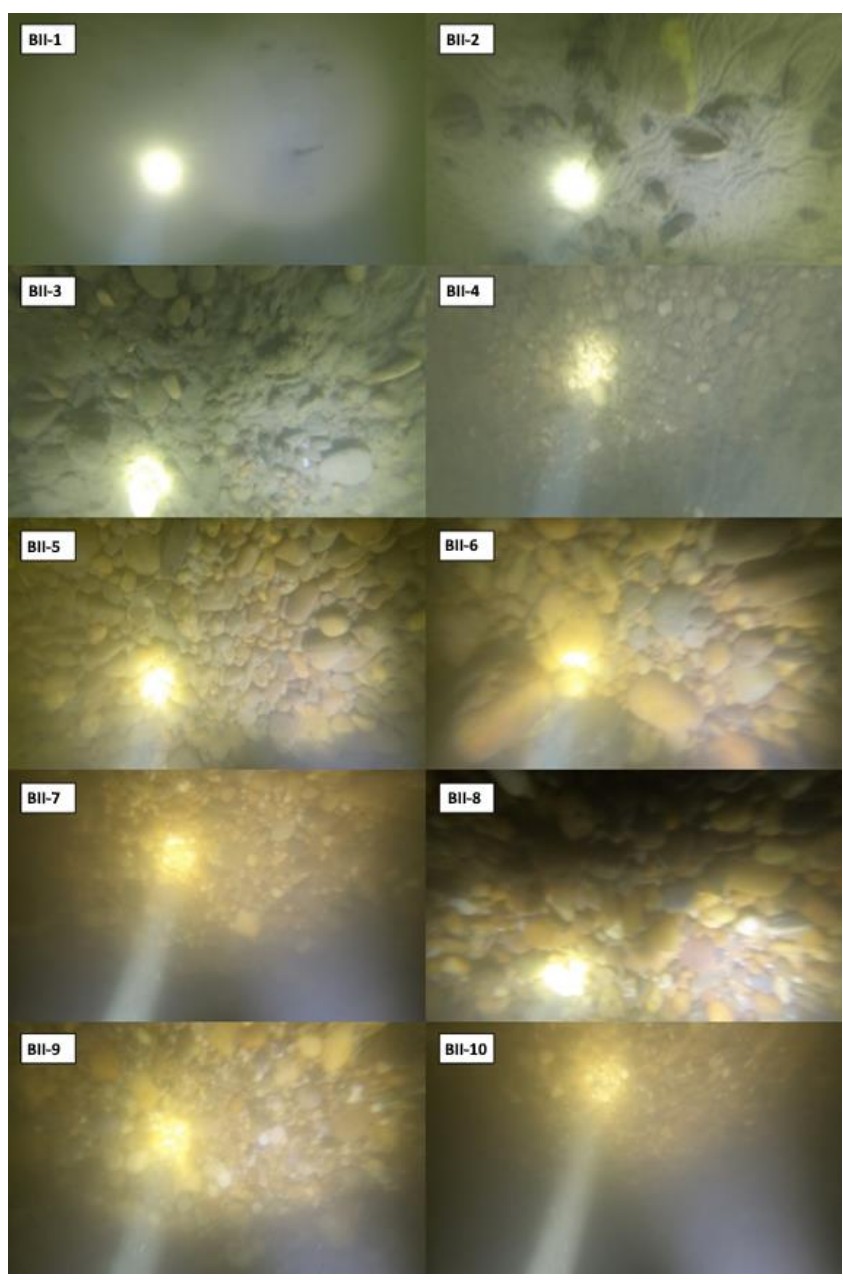

**Figure B1: Riverbed video images at the sampling points in Section B - II.**

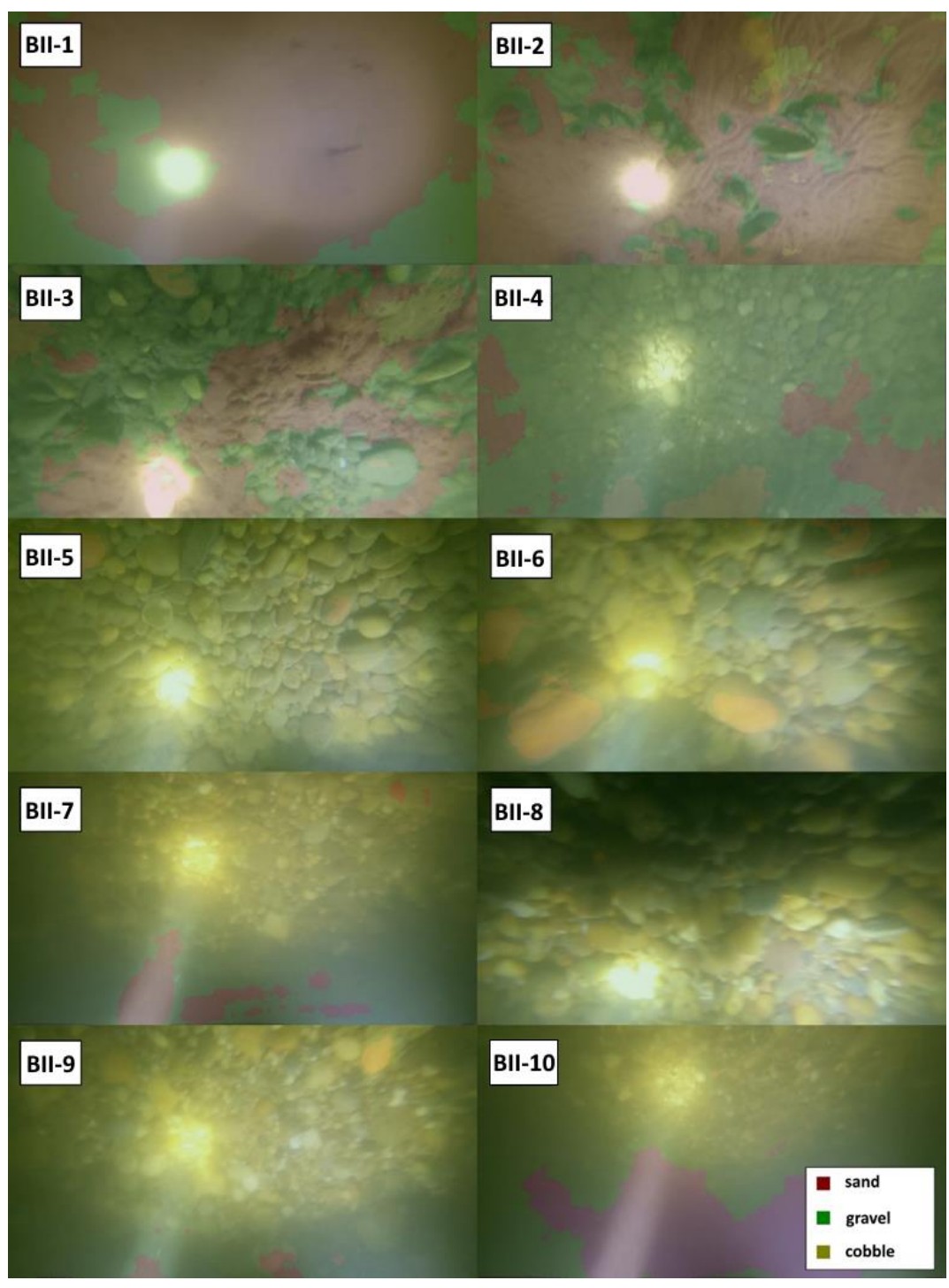


**Figure B2: Riverbed video images overlapped with their raw, DL detection result, at the sampling points in Section B**
**- II.**

    **Appendix C Site A - Section A – I**

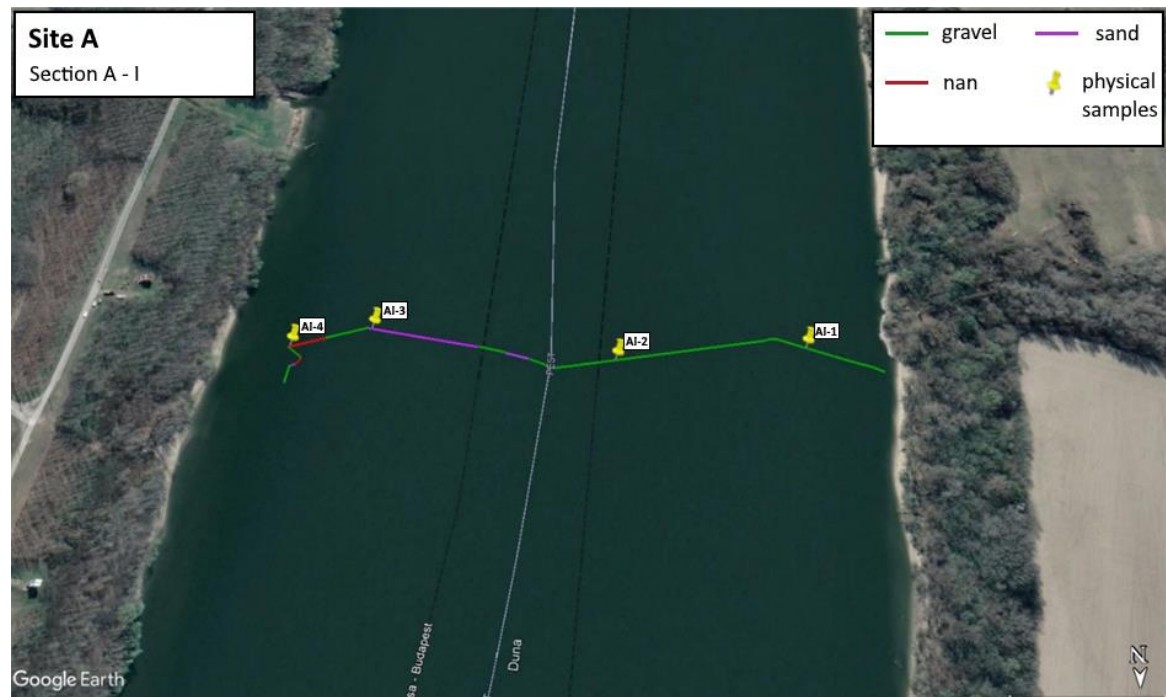

**Figure C1: The path of the vessel and camera in Section A - I, Site A. The polyline is coloured based on the sediment**
**seen during visual evaluation of the video. Yellow markers are the locations of physical bed material samplings. (Map**
**created with Google Earth Pro)**

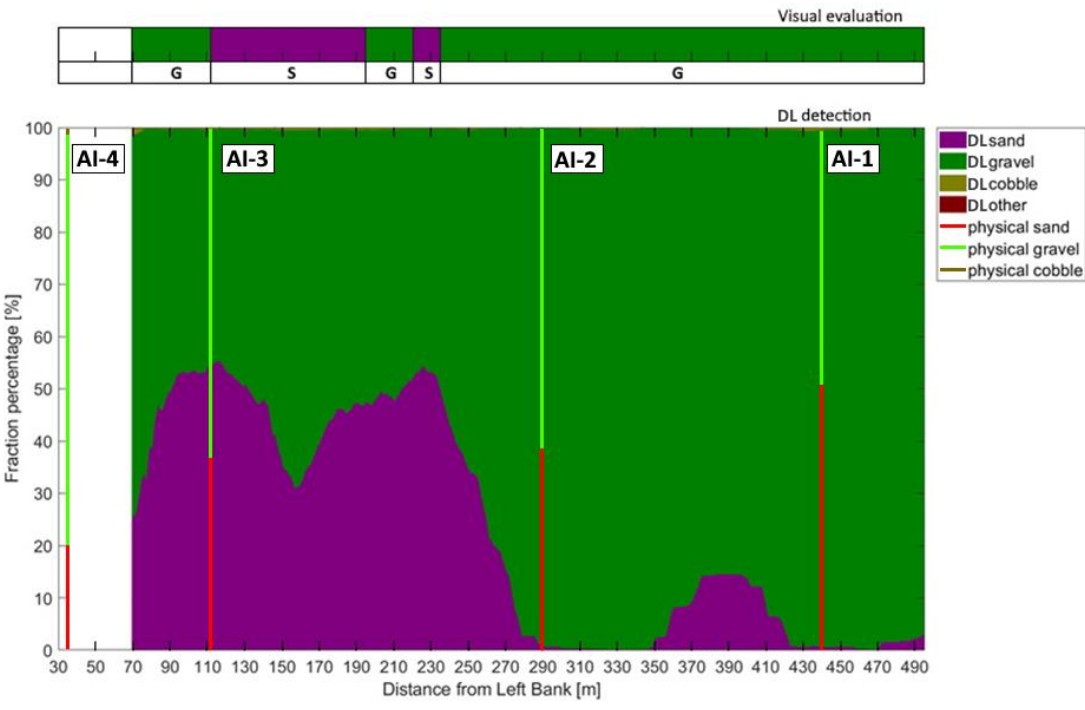

**Figure C2: Sediment fraction percentages in Section A - I, recognised by the AI. The visual evaluation included two**
**classes: gravel – G, sand – S). The fractions of the physical samples are shown as verticals.**

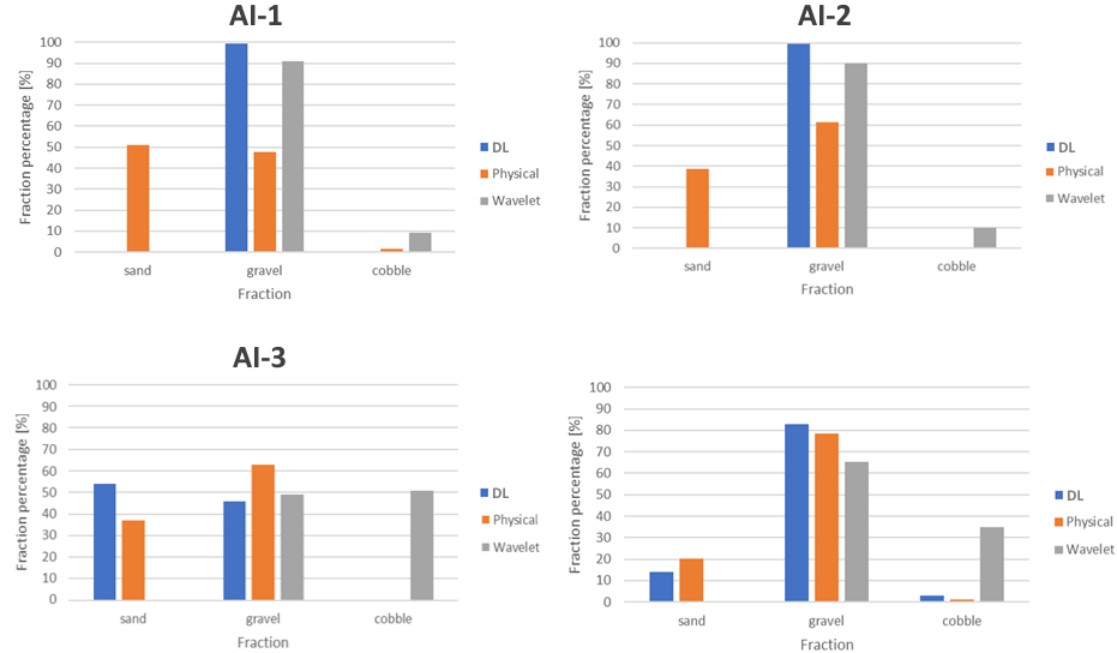

Figure C3: Comparison of sediment fraction % at the sampling locations from the moving-averaged DL detection, conventional sieving and the wavelet-based image processing method. Section A - I.

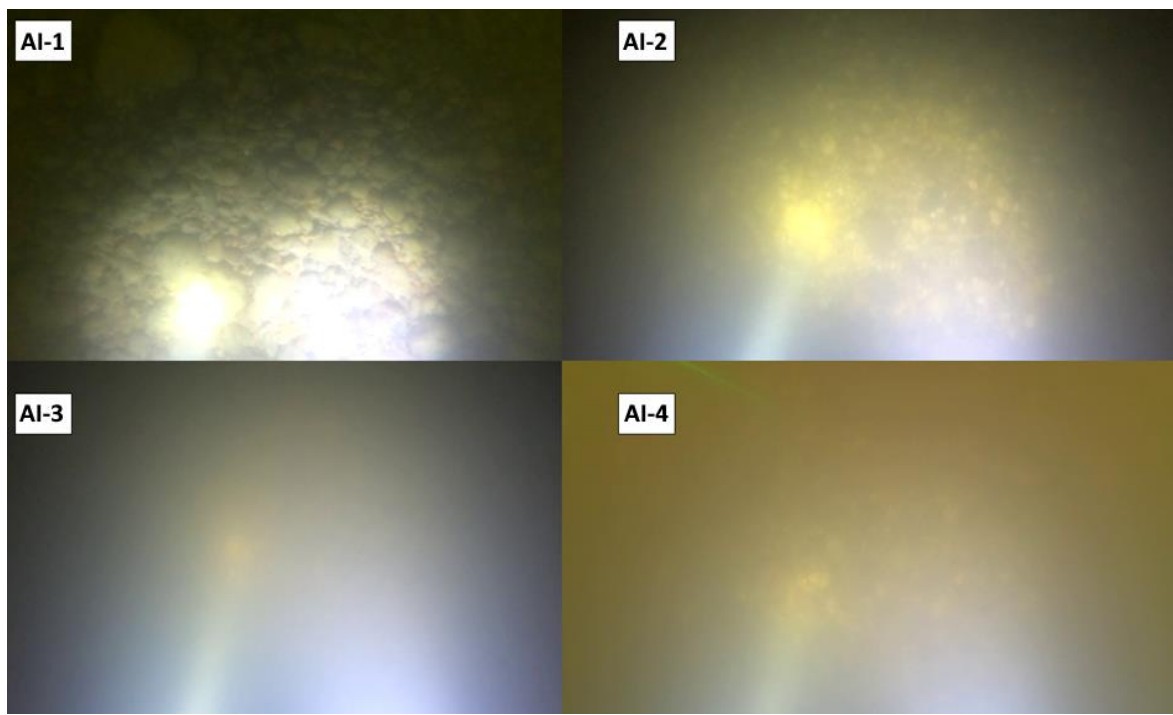

Figure C4: Riverbed video images at the sampling points in Section A - I.

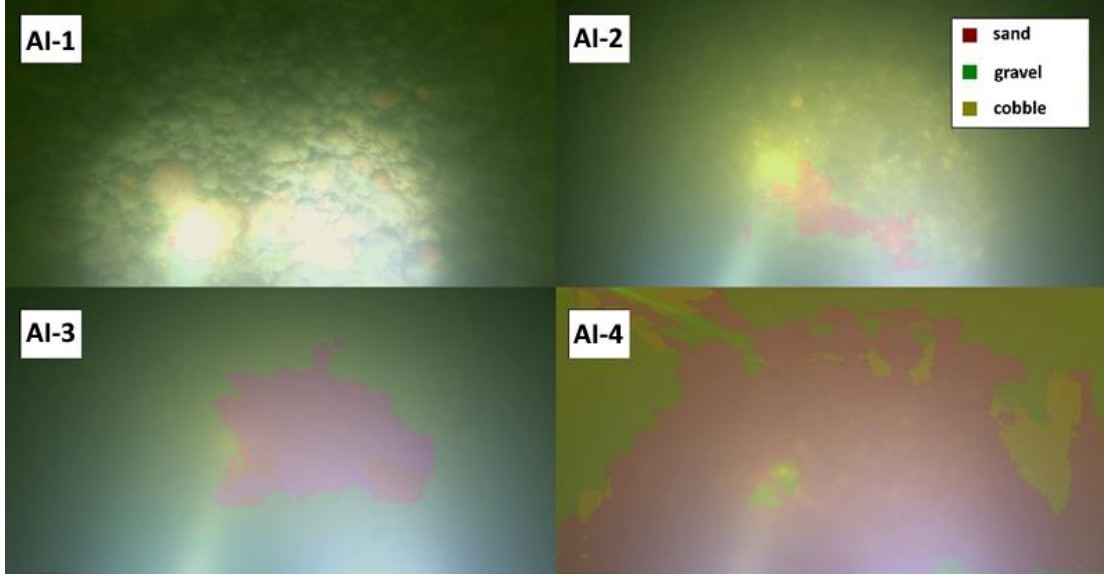


**Figure C5: Riverbed video images overlapped with their raw, DL detection result, at the sampling points in Section A**
**- I.**

 **Appendix D Site A – Section A - III**

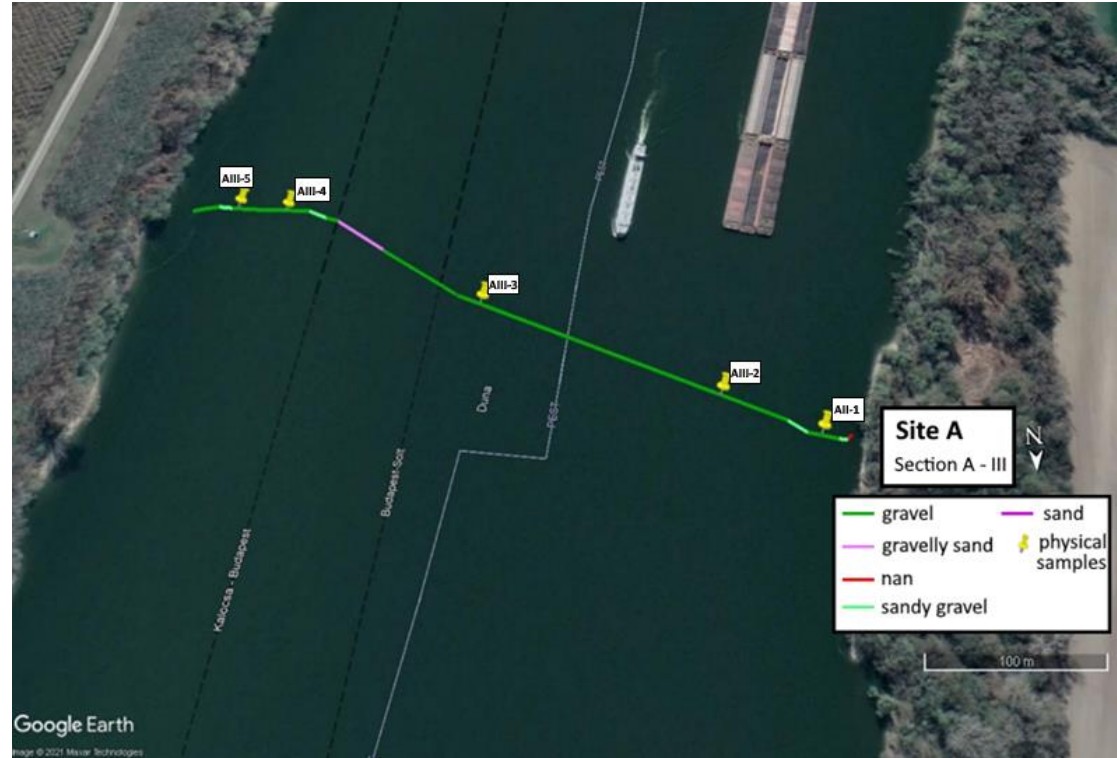

**Figure D1: The path of the vessel and camera in Section A - III, Site A. The polyline is coloured based on the sediment**
**seen during visual evaluation of the video. Yellow markers are the locations of physical bed material samplings. (Map**
**created with Google Earth Pro)**

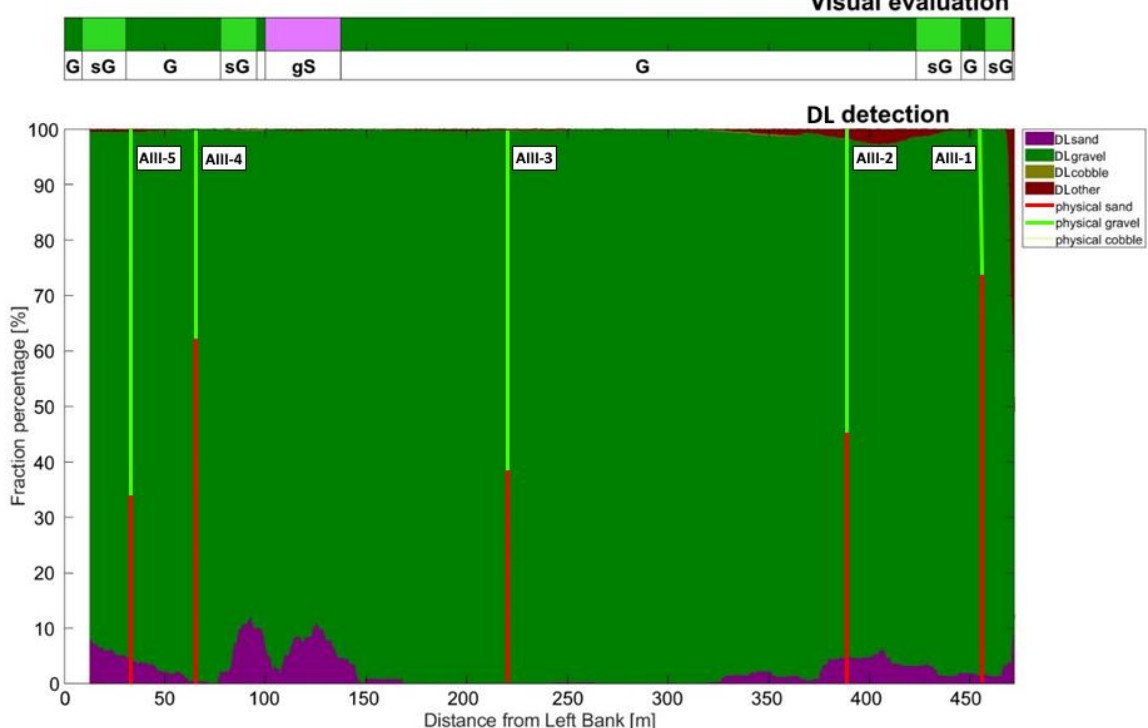

**Figure D2: Sediment fraction percentages in Section A - III, recognised by the AI. The visual evaluation included three**
**classes: gravel – G, sandy gravel – sG, gravelly sand - gS). The fractions of the physical samples are shown as verticals.**

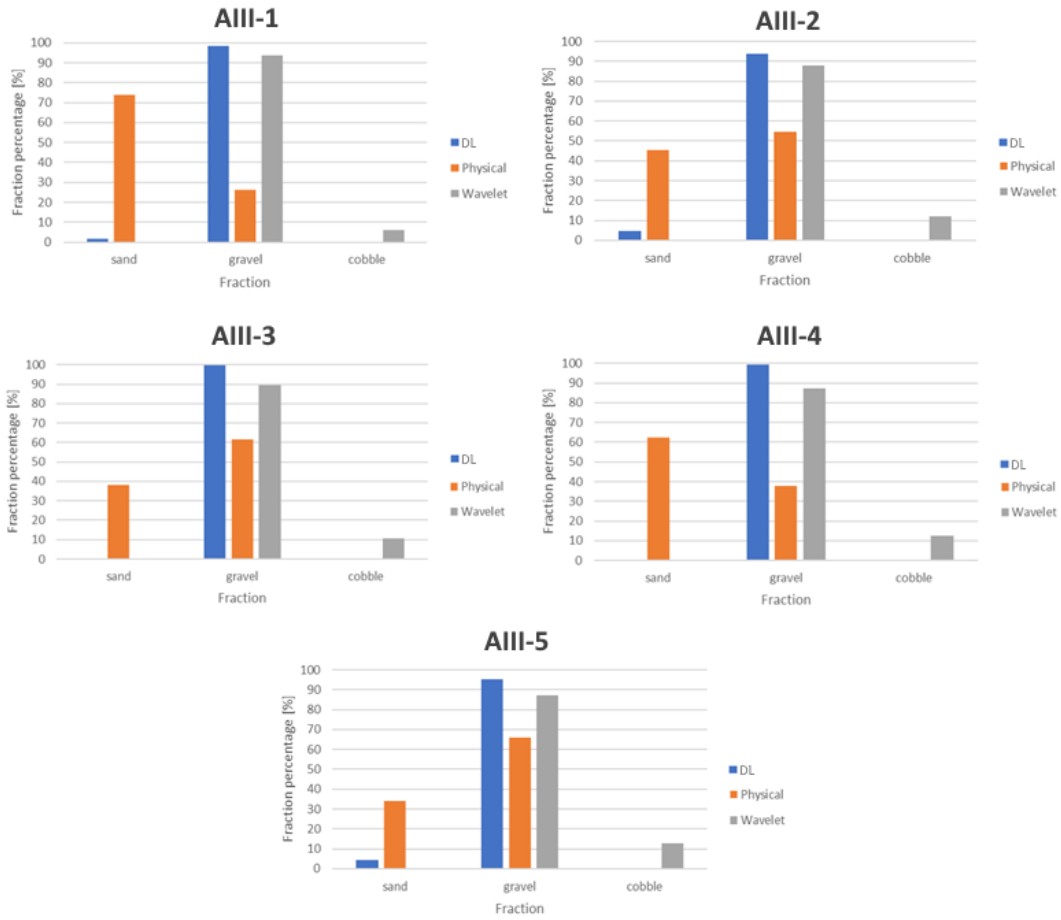

**Figure D3: Comparison of sediment fraction % at the sampling locations from the moving-averaged DL detection,**
**conventional sieving and the wavelet-based image processing method. Section A - III.**

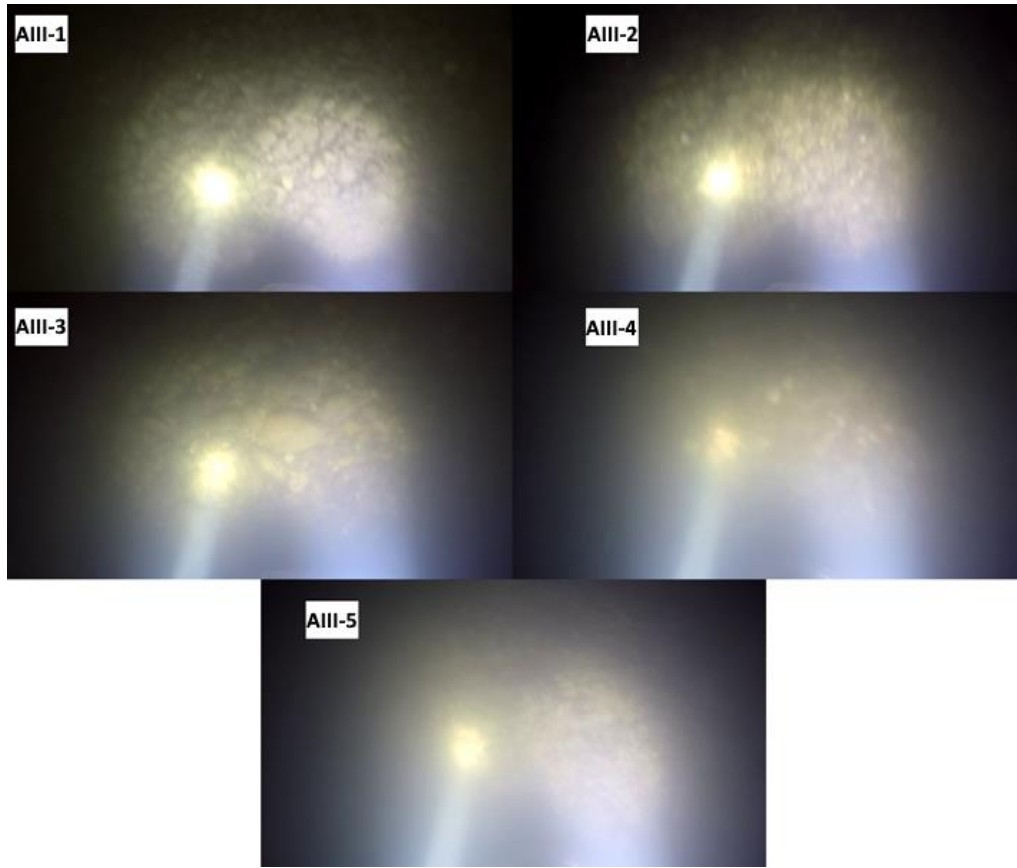

**Figure D4: Riverbed video images at the sampling points in Section A - III.**

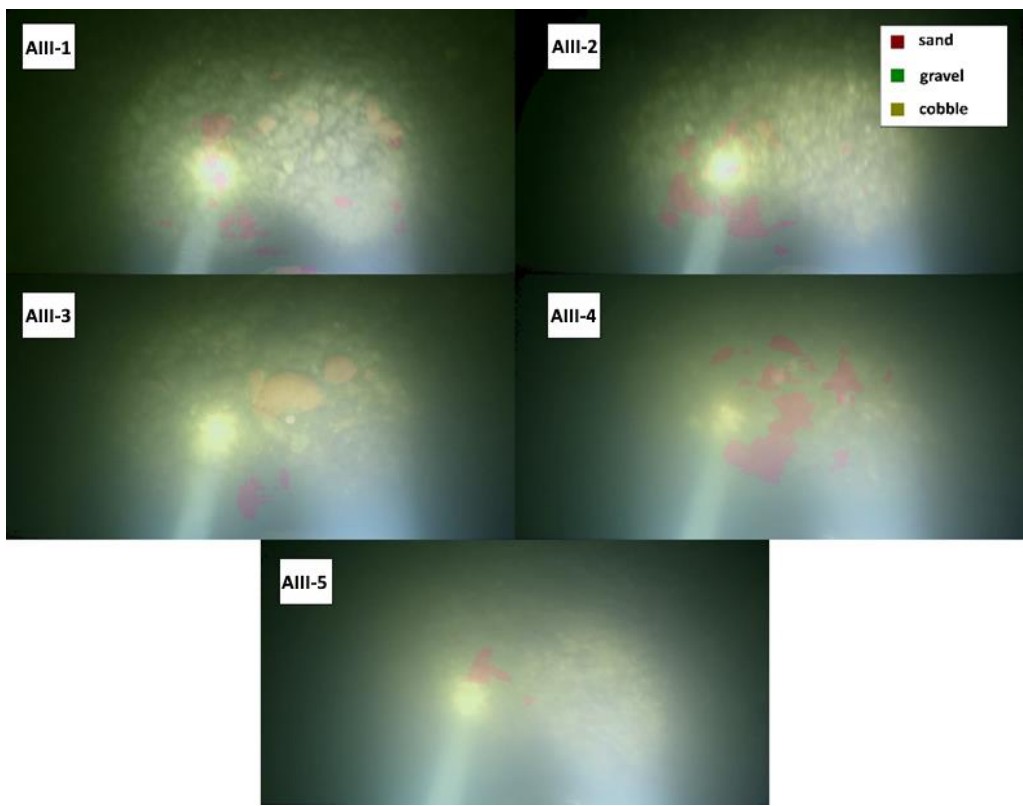

**Figure D5: Riverbed video images overlapped with their raw, DL detection result, at the sampling points in Section A**
**- III.**

 **Appendix E Site B – Section B - I**

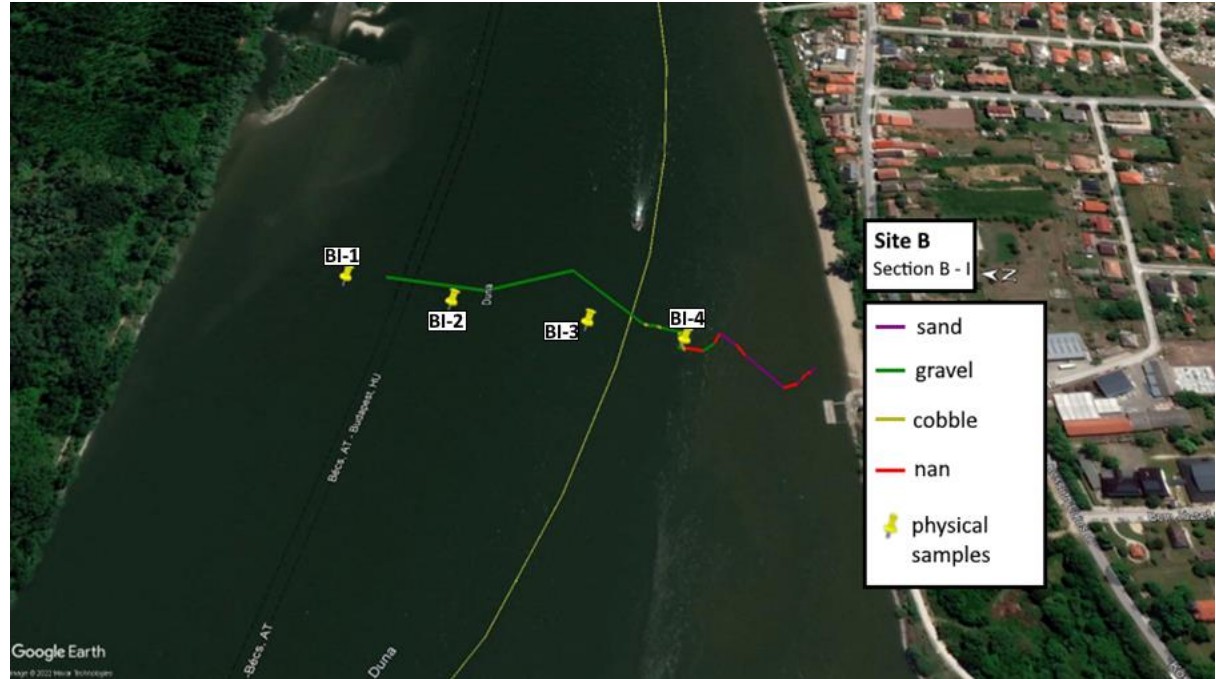

 **Figure E1: The path of the vessel and camera in Section B - I, Site B. The polyline is coloured based on the sediment seen during visual evaluation of the video. Yellow markers are the locations of physical bed material samplings. (Map created with Google Earth Pro)**

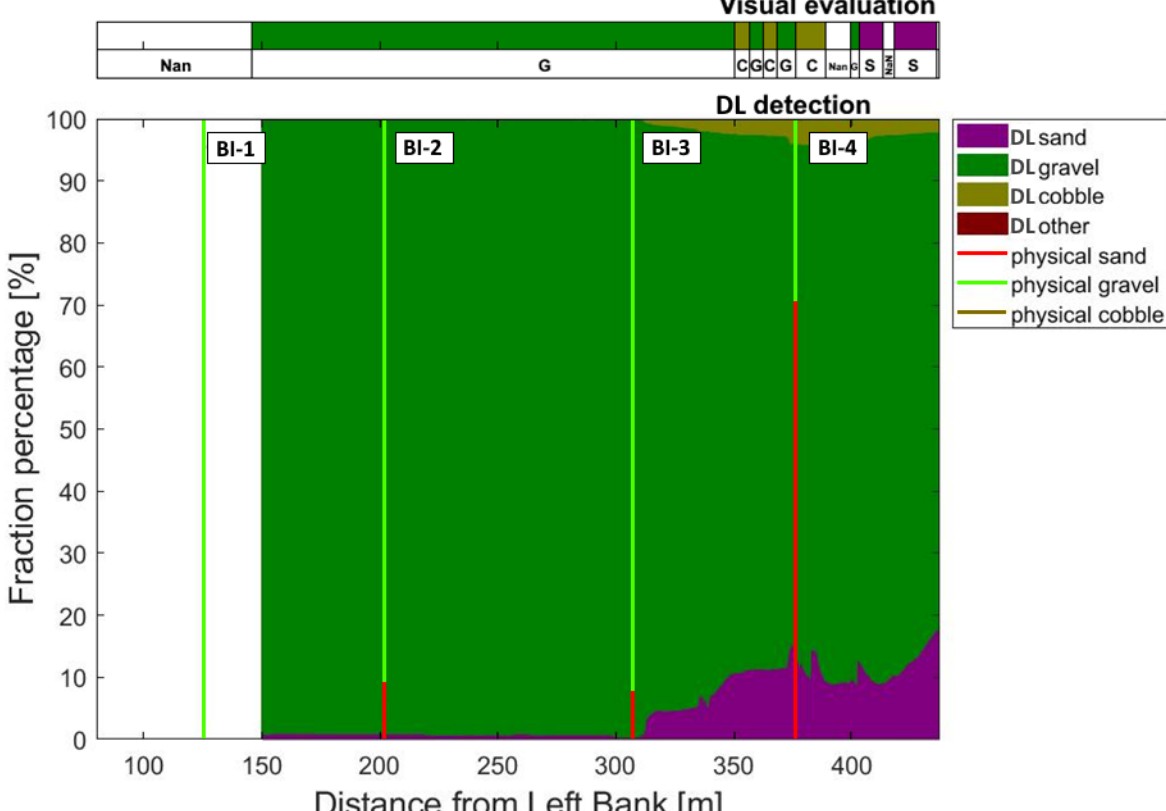

 **Figure E2: Sediment fraction percentages in Section B - I, recognised by the AI. The visual evaluation included two classes: gravel – G, sand – S). The fractions of the physical samples are shown as verticals.**

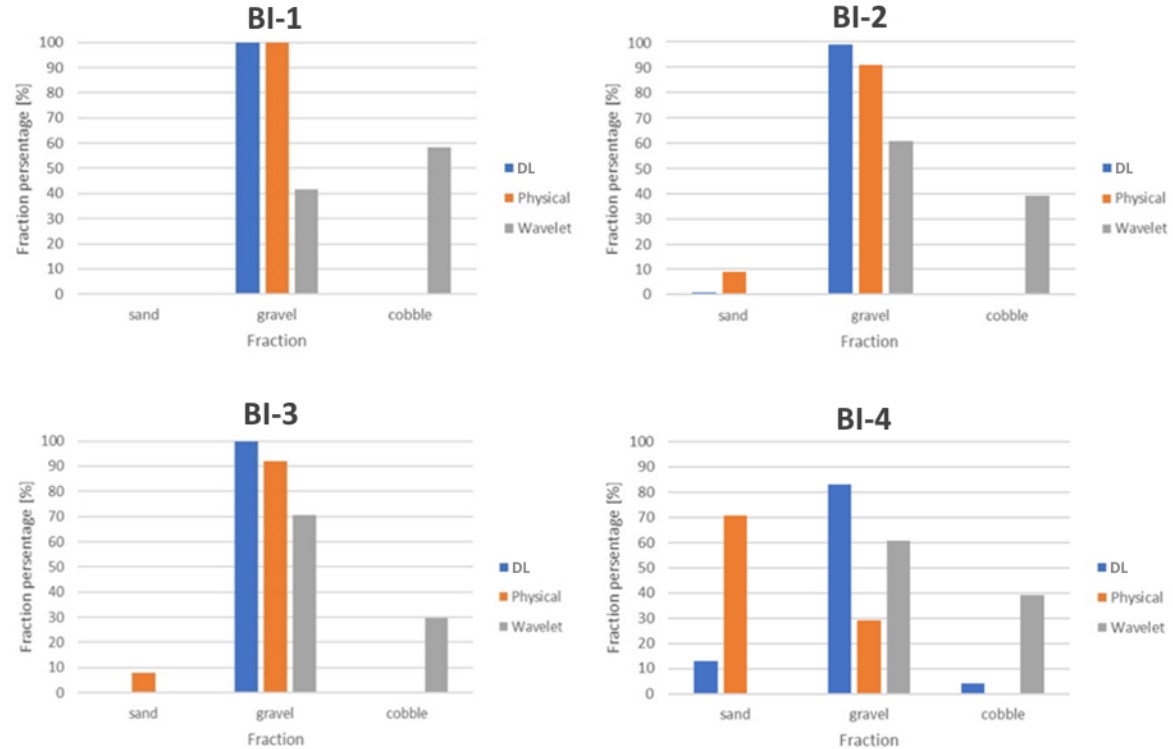

**1141** **Figure E3: Comparison of sediment fraction % at the sampling locations from the moving-averaged DL detection,**
**1142** **conventional sieving and the wavelet-based image processing method. Section B - I.**

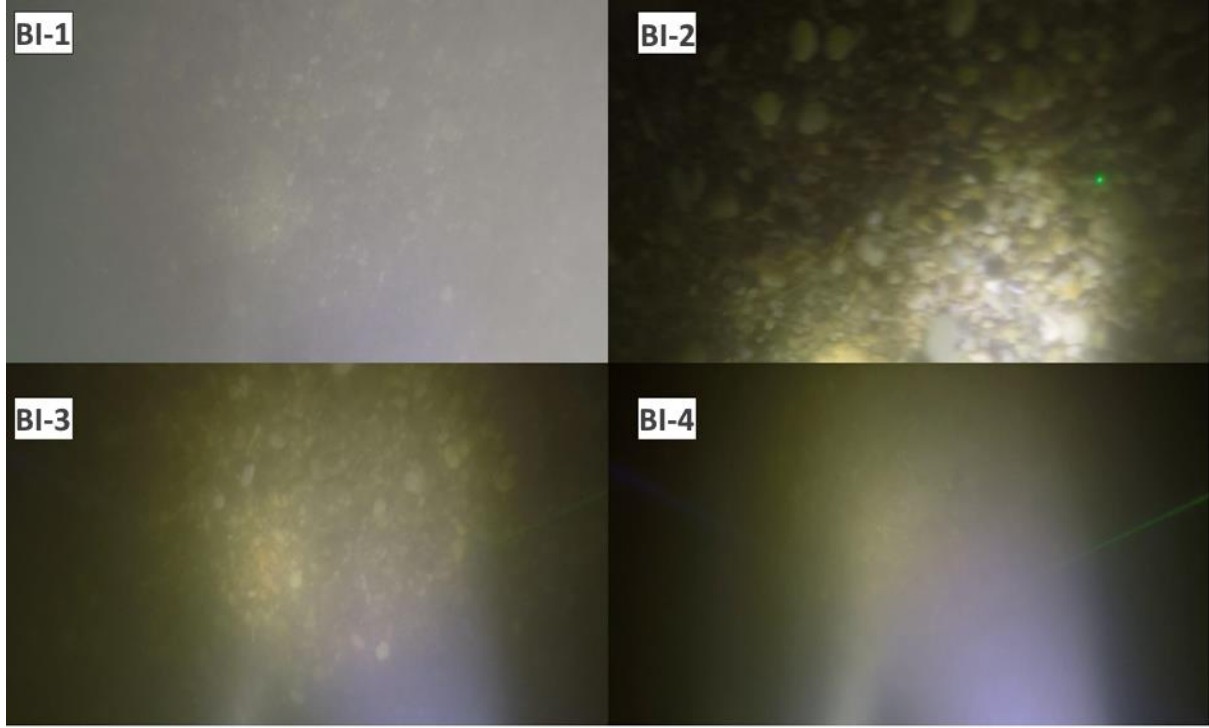

**1143**
**1144** **Figure E4: Riverbed video images at the sampling points in Section B - I.**

**1145**

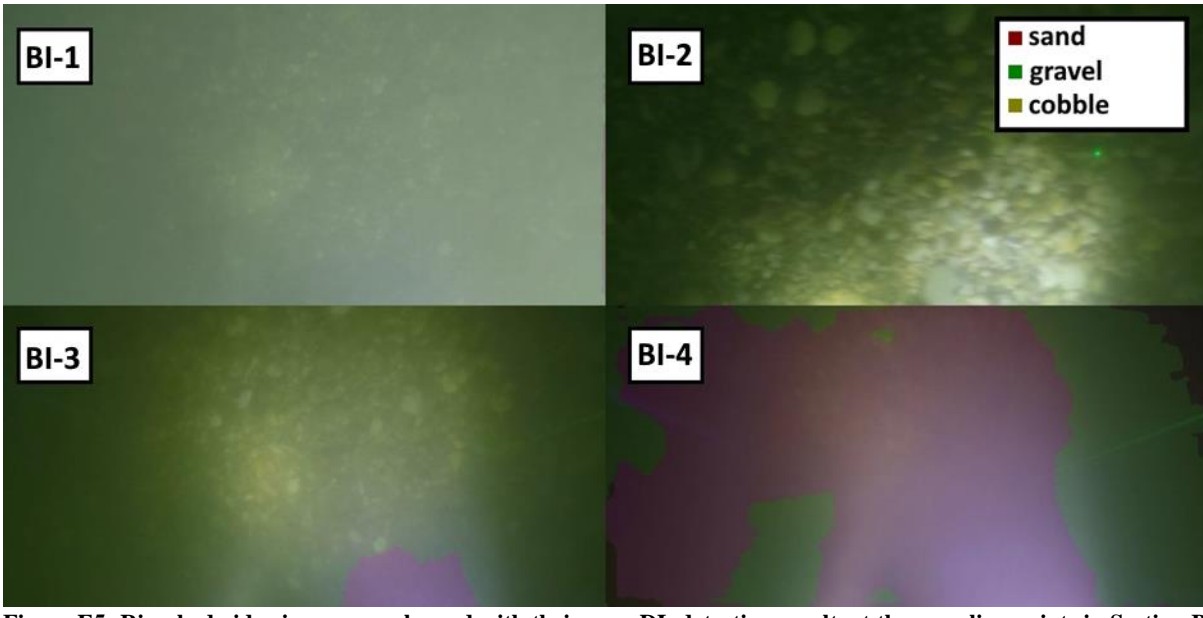

**Figure E5: Riverbed video images overlapped with their raw, DL detection result, at the sampling points in Section B - I.**

**Appendix F**

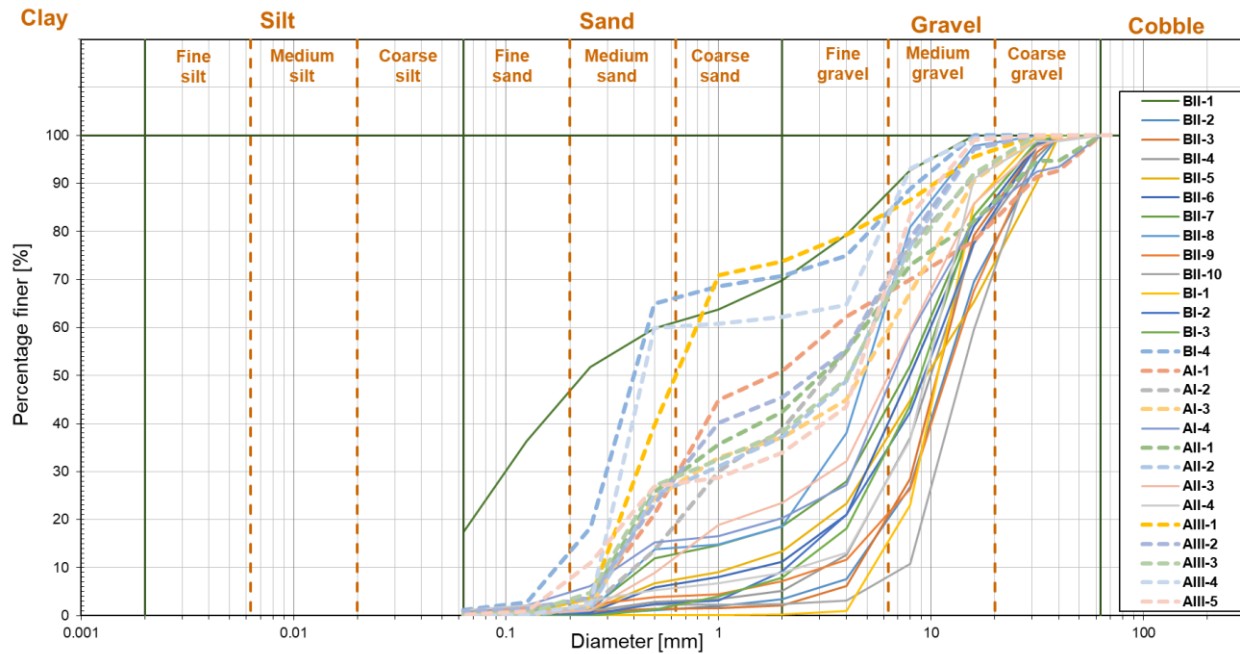

**Figure F1: Grainsize distribution curves of the 27 sieved physical samples. 11 curves categorised as Outlier Type-A are showcased with dashed lines. The shapes of these curves are representing bimodal (gap graded) sediment distributions, which typically refers to bed armouring (i.e., excess of a certain particle size, a coarser surface layer protects a finer subsurface layer from being washed away). Hence, analysing images of the surface layer could not represent these complex distributions inherently.**