# Peer review of "Automated riverbed composition analysis using Deep Learning"

_Earth Surface Dynamics, 2022_

## Author Comment (AC1)

Dear Reviewer,

We have structured our response in a Q&A form. In each case, your Question/Comment comes first, and then it is followed by our Answer.

**1. General remarks**

**Question/Comment from Reviewer:** Paper is in general well structured, and methods are clearly presented. Results of the analyses with proper discussion could be improved.

Authors have done significant work, processing large volumes of the video data manually, and using wavelet and AI methods. Since extensive data preparation and analyses are required for these methods, authors have decided to present the results as bar plots over the sampling points or over the boat transect. This way, majority of the data processing remains hidden and readers have little insight into the detailed functioning of each automated method. The section of the paper dedicated to results and comparison of the methods (4.2.) contains only 4 paragraphs. In my opinion, the results should be discussed in more detail as they are key for validation of your approach.

I suggest that authors additionally present the complete results of the data processing (on an image-by-image scale) in form of frequency distribution curve. This way more metrics about noise in the results would be given to complement the final result.

Overall, presentation of the results (figures and graphs) could be upgraded to match the quality of the conducted research.

**Answer:** *Thank you for your comment! We followed your suggestion.*

*Primarily:*

- *We have introduced Fig. 22 and Fig 23. Explanation and reasoning for these new figures are also given: Line 21 – 24 (Abstract), Line 552 – 575; Line 642 – 645.*

*Secondarily:*

- *To better show the performance of the AI algorithm we have included Fig. 8 (i.e., raw AI results of a survey in one of the sections; before applying moving-average). We have also included explanation on this: Line 400 – 413.*
- *Even though due to the example of Fig. 8, we use the moving-averaged results of the AI when compared to wavelet and physical sieving, we decided to add the AI detection for the specific images taken in the sampling points (basically the AI values from before moving-average in the given point). See: Fig. 14, Fig. 21, Fig. A5, Fig. B5, Fig. C5. We believe these would give extra insight on the current performance and shortcomings of the presented method and help the readers.*

A: *Thank you for your comment! We have introduced visual supplements by inserting Fig. 10, Fig. 12 and Fig. 19. These include bed armouring, different sediment patches and the wavelet method for both cases. We intended to choose the most telling and prominent sample points for each case.*

Q: Authors propose their setup to be moored from the boat on a line and weighted down, assuming that the setup remains horizontal throughout the deployment. However, depending on the line length and drag influence on the setup geometry (which is highly irregular), it is inevitable that setup will be tilted during measurement. Authors haven't presented any details on how this affects the video data, which might be crucial since there are no reference points on the bed.

A: *Thank you for your comment! We have added some explanation in the manuscript, in ch. 4.3. (Line 592 - 606).*

Q: Lastly, authors have addressed implementation challenges in the section 4.3., which will be helpful in further development of this approach. However, challenges they have detected are result of the selected approach, not universal and therefore need to be addressed accordingly. E.g. vessel speed lower than 0.2 m/s is claimed not feasible for straight transect over the river. Since the proposed method aims to cover broader river section and result in 2D maps, why is it important to transect the river? Maybe longitudinal approach would allow for lower velocities while covering the same area? Implementation challenges should be addressed universally and shortcomings of your selected approach (boat type, camera type, illumination, transects, etc.) should be provided with discussion of the data. This is one of the reasons that this section contains repetitive text from earlier, and again some of the findings are repeated in the following section 4.4. Novelty and future work.

A: *Thank you for your comment! Relating to the vessel speed and longitudinal survey paths, we have added some explanation in the manuscript, in ch. 4.3. (Line 580 – 592). Regarding the repetition of illumination, we have removed this part from ch. 4.3. and inserted it at ch. 4.2, where the problem of illumination first came up and is visually supplemented (Line 517 - 523). We did our best to remove/separate the specific challenges from ch. 4.3. and added them to the section where the setup is introduced, ch. 3.2. (Line 221 – 231).*

**2. Specific remarks**

Q: Keywords are relatively general and do not offer information of specific contents of the paper. I suggest to drop the keywords "rivers", "sedimentology", "underwater" and "mapping", and use the following keywords for distinction instead: "riverbed texture" and "underwater mapping" (or something along these lines).

A: *Thank you for your comment! We complied with your suggestion and modified the keywords to:*

*riverbed texture, underwater mapping, sediment classes, Artificial Intelligence, Deep Learning, image-based.*

Q: There are several issues with referencing approach used in the paper:

Reference "Török et al., 2017" is not appropriate to be representative of bed armour development since the paper focuses on CFD modelling.

A: *Thank you for your comment! We have replaced it with Ferdowski et al., 2017. (Ferdowsi, B., Ortiz, C. P., Houssais, M., & Jerolmack, D. J. (2017). Riverbed armouring as a granular segregation phenomenon. Nature Communications 2017 8:1, 8(1), 1–10. https://doi.org/10.1038/s41467-017-01681-3)*

Q: On ln43 several references from the same author are used where one would be sufficient (Church et al., 1987; Wolcott and Church, 1991; Rice and Church, 1998; USDA, 2007)

A: *Thank you for your comment! We have removed Church et al., 1987 and Rice and Church, 1998.*

Q: On more occasions there are several references grouped together, masking specific relevance for each of them - see ln48, ln73, ln83, ln86, ln88, etc. I suggest that in cases where all references support the same statement select the ones most representative, and in cases where more statements are given you connect the reference with each of them (e.g. ln86)

A: *Thank you for your comment! We have removed several references and left the most representative ones.*

Q: Reference "Kellerhals and Bray, 1971" is used 3 times in the paper, always grouped with several others and twice with Adams. I suggest that you use it where it is most appropriate (ln50) and drop it for general statements (ln40 and ln82)

A: *Thank you for your comment! We have followed your suggestions.*

Q: In the goal of the paper contribution is highlighted as "…through improved (continuous, quick, covering larger areas) data collection.". I would suggest that you rephrase this into "…through more extensive data collection." Since it is hard to argue that method is:

…continuous (data is collected over single in situ survey)

…quick (single point collection is quicker, both for collection and processing)

…covering large area (what is large area, and does this conflict with two previous advantages)

I suggest that these advantages authors address in the discussion and drop from the goal itself since they are not straightforward.

A: *Thank you for your comment! We have followed your suggestions.*

Q: Considering the size of the paper and volume of the work conducted, the goal of the paper is a bit short. I suggest that you expand the goal of the paper to reflect the specific contribution to the field.

A: *Thank you for your comment! We have followed your suggestion and extended the description of our goal in ch. 1. (Line 152 – 161).*

Q: Methodology section briefly describes the three locations of Danube River where data was collected, introducing flow rate data, SSC, etc. This data might be useful for someone familiar with the river, which most of the readers won't be. Please put the presented data in context – e.g. provide complementary duration data, long-term average data, etc. which would allow estimation of conditions under which surveys were performed (low flow, average flow, flood).

A: Thank you for your suggestion! We have followed your suggestion and added $Q_{1\%}$. We also attempted to clarify the Table.

Q: Part of the Methodology focusing on the equipment lacks information that would help understand the data quality, in the context of the maintaining the setup distance from the bed. Please expand the current description with the data about the desired height above the bed and what does it depend on (supposably illumination of the FOV). Similarly, you initially state that size-reference wasn't used in the images (ln253), and after offer contrary statement that laser pointers were used to provide scale (ln294). Probably laser pointers do not offer constant distance due to the bed irregularities, but clarification of the way they were used would be helpful.

A: *Thank you for your comment! We have expanded the camera - riverbed distance related description: (Line 580 – 592).*

*In ln173, it was originally said the training was unscaled for the AI. Ln371 - 384 was about the wavelet method, which inherently requires scaling. The wavelet method was only applied in the physical sampling points, so the laser were only used in these points. As we had some problems with them, we did not use the lasers constantly, during the transversal surveys, only when we reached over a given physical sampling point. We have clarified these statements and updated the manuscript (Line 341 – 346).*

Q: In the Section 4.1, explaining the training of Deep Learning authors present example of erroneous particle detection of the user. Although this is good and informative example, in my opinion it shouldn't be presented as "ground truth" and rather included as sidenote explaining that training data needs to be carefully selected since you noticed errors in user judgement (n.b., who was the user – one of the authors or trained personnel?)

A: *Thank you for your comment! We have modified the Figure and separated the case of erroneous detection, along with its description. Fig. 5 and Fig. 6, Line 366 – 372.*

*N.b. The user was trained personnel. We have now also included this in the text. Line 296.*

3. Technical remarks

Q: On ln 31 "fluvial navigation" -> more appropriate would be "fairway placement", but the connection is loose so I suggest that you replace it with use more relevant to the grain sire instead.

A: *Thank you for your comment! We have followed your suggestion and looked for the most appropriate expression. We have decided with "inland waterway transport".*

Q: On ln 31 "riverbed structure" -> unclear, does this represent morphology?

A: *Thank you for your comment! We have corrected the sentence to:*

*"Knowledge of riverbed () morphology and () sediment composition is therefore of major importance in river hydromorphology."*

Q: "Riverbed" and "river bed" are used interchangeably throughout the manuscript. Please proofread the manuscript.

A: *Thank you for your comment! We have corrected it.*

Q: Goal (aim) of the paper is combined with Introduction, making it indistinctive. Please separate the aim of the paper into separate paragraph.

A: *Thank you for your comment! We have followed your suggestion! Line 152 – 161.*

Q: Figure 1 is very simple and lacks details (Danube is not highlighted and therefore indistinctive from other rivers in the figure).

A: *Thank you for your comment! We have updated the figure and highlighted the Hungarian Section of the Danube.*

Q: Can the background orthophoto data be added on the Figures 2 and 3?

A: *Thank you for your comment! We have followed your suggestion! We also merged them into one figure.*

Q: Names of the sections (transects) are very hard to follow since they do not follow any logical order (I suppose they do for the authors, but I suggest that you rename them to achieve clarity for the readers).

A: *Thank you for your comment! We have followed your suggestion!*

Q: "Streamlined weight" -> isokinetic suspended sediment sampler?

A: *Thank you for your comment! Yes. We have added the explanation:*

> *" (...) streamlined weight (originally used as an isokinetic suspended sediment sampler)  (...)".*
> *Line 211 – 212.*

Q: Methodology section would benefit from added flow chart (process chart) since video decomposition and enhancement is carried out through several steps.

A: *Thank you for your comment! We have added a flow chart in Fig. 4.*

Citation: https://doi.org/10.5194/esurf-2022-56-RC1

---

## Author Comment (AC2)

Dear Reviewer,

We have structured our response in a Q&A form. In each case, your Question/Comment comes first, and then it is followed by our Answer.

Question/comment from Reviewer: In general, the manuscript is well written and structured, although I recommend some small re-structuring at the beginning. The methods are mostly explained in a clear way, however, the way of presenting the results have to be modified to improve the scientific quality of the manuscript before publishing.

Answer: *Thank you for your comment! We followed your suggestion.*

    *Primarily:*

- *We have introduced Fig. 22 and Fig 23. Explanation and reasoning for these new figures are also given: Line 21 – 24 (Abstract), Line 552 – 575; Line 642 – 645.*

    *Secondarily:*

- *To better show the performance of the AI algorithm we have included Fig. 8 (i.e., raw AI results of a survey in one of the sections; before applying moving-average). We have also included explanation on this: Line 400 – 413.*
- *Even though due to the example of Fig. 8, we use the moving-averaged results of the AI when compared to wavelet and physical sieving, we decided to add the AI detection for the specific images taken in the sampling points (basically the AI values from before moving-average in the given point). See: Fig. 14, Fig. 21, Fig. A5, Fig. B5, Fig. C5. We believe these would give extra insight on the current performance and shortcomings of the presented method and help the readers.*

1. General comments:

Q: The abstract contains a brief description of the methods and its advantages compared to conventional measuring procedures and it is mentioned what is included in the manuscript without naming results. Basically, no results are presented in the abstract. E.g., how many sites showed an successful performance etc. One reason might be the missing quantitative evaluation of the performance of the new technique compared to other techniques (see also comment below).

A: *Thank you for your comment! We have introduced Fig. 22 and Fig 23. Explanation and reasoning for these new figures are also given: Line 21 – 24 (Abstract), Line 552 – 575; Line 642 – 645.*

Q: Regarding the structuring the authors choose to write introduction and literature review separately leading to the issue that the shortcomings and restrictions of alternative measuring concepts are included in the chapter of the literature review but the objective and goals of this manuscript is included in the introduction. This does not read straightforward and I recommend to rethink the structuring in this context.

A: *Thank you for your comment! We agree so we have updated and restructured Ch. 1. accordingly.*

Q: The authors present a new AI-approach and used the River Danube as study site. Did the authors make a proof-of-concept or some experiments to test the AI approach under easier boundary conditions? From my perspective, the new AI approach should be first evaluated for easy conditions (maybe shallow and clear water, laboratory conditions, manually selected surface compositions), before applying it at much more challenging conditions.

A: *Thank you for your comment! Yes, we have updated the manuscript and now shortly mention Benkő et al., 2020 as our "proof-of-concept" study, in ch. 3.3., Line 260 – 263.*

*Note for the Reviewer: In Benkő et al., 2020 the same AI architecture was applied for analysing drone videos of a dry riverbed. The drone was easily maintaining constant distance from the bed during its flight, enabling scaling of the images. Furthermore, the visibility conditions were far more friendly (compared to underwater conditions), as the measurements were carried out during clear and sunny weather.*

Q: Another question that can/should have been discussed is about the representativity of the evaluated areas. Similar to the weight of a sediment sample an area can be specified to be sure that it is representative for the river. Well, I see that this manuscript is mainly to present the method but the representativity aspect should at least be discussed and be mentioned in the outlook.

A: *Thank you for your comment! Indeed, the question of representativity needs to be discussed in future work. In this manuscript, as you mention, the exploration of field application and performance of the AI was our main goal. As such, we included some thoughts on this as outlook, in Ch. 4.4, Line 697 – 704.*

Q: My main concern is the presentation of the results. Although I like the way of showing the results (Figure 5-7), it is too less from a scientific point of view. The manuscript include only a fraction of the verification and validation process. I recommend that authors think about a way to include all results in the paper in a quantitative way. Although the mathematical tools (pixel accuracy, IoU) might not be perfect instruments for the validation, they provide valuable information that can be calculated for the entire datasets and results can be presented e.g. in box plots. This would include the entire work that was done and provide much more information compared to the currently presented results. I also recommend for Figures 8,13 to calculate the differences between the AI approach and the others (for all datasets) and again…the differences can be plotted e.g. as box plots. Overall, this means that a more quantitative way of the verification should be achieved.

A: *Thank you for your comment! We have introduced Fig. 22 and Fig 23. Explanation and reasoning for these new figures are also given: Line 21 – 24 (Abstract), Line 552 – 575; Line 642 – 645.*

2. Specific comments:

Q: Abstract Line 21-22:

The last sentence of the abstract represents the aim/goal/purpose of this manuscript? I suggest shifting the purpose to a more appropriate place.

A: *Thank you for your comment! We have removed the last sentence.*

Q: Keywords Line 23:

From my perspective the selected keywords are not descriptive enough for the presented work (too general words/terms)

A: A: *Thank you for your comment! We have modified and changed the keywords.*

Q: Introduction Line 55-58:

These lines describe the goal of the paper. However, the goal is not described precisely. The authors mention to "introduce a Deep Learning-based technique" without writing for what. My recommendation is to rephrase the goal/objective of the manuscript e.g. by formulating a hypothesis.

A: *Thank you for your comment! We have followed your suggestion and extended the description of our goal in ch. 1. (Line 152 – 161).*

Q: Introduction Line 56, 64, 183 ff

I am not sure, if the term "paper" is sufficient or the correct term for a scientific article (maybe too colloquial?), of if "manuscript" would be the better term (I am not a native speaker...).

A: *Thank you for your comment! We have replaced the word with "manuscript".*

Q: Literature Review, Line 67

"One group of the surrogate approaches are acoustic methods" instead of "One group of the surrogate approaches is the acoustic methods"

A: *Thank you for your comment! We have corrected it.*

Q: Literature Review, Line 106-107

This last sentence does not fit into the listing because it describes the effect of the limitation. However, I do not agree with this last sentence in general. Although image-based analyses might be time-consuming in post-processing, they are very helpful and quick in the field, e.g. to collect sediment data for several kilometres of a river that can also be seen as quick and continuous measurement. From my perspective, the question of "quick" depends on the goal of a survey…

A: *Thank you for your comment! We have removed the last sentence.*

Q: Literature Review, Line 116-117

How can get information about a GSD without detecting the size of particles? Please clarify or rephrase.

A: *Thank you for your comment! We rephrased it to: "For instance, Rozniak et al. (2019) developed an algorithm for gravel-bed rivers, performing textural analysis. With this approach, information is not gained on individual grains (e.g., their individual shape and position), but rather the general grain size distribution (GSD) of the whole images". Line 107 - 109.*

Q: Literature Review, Line 162-168

The article structure using an introduction (with specification of goals), then an literature review with shortcoming of previous method and benefits of the presented new method is a bit strange. I recommend to include the literature review into the introduction (most common way) and then objectives/aims/goals can be specified based on shortcoming of previous methods and/or novelty/benefits of the new method.

A: *Thank you for your comment! We agree so we have updated and restructured Ch. 1. accordingly.*

Q: Figure 1

The formulation of the caption could be improved.

A: *Thank you for your comment! We have modified the caption.*

Q: Figure 2, Figure 3:

The captions should include that the figure is showing the bathymetry of Site A with three measurement transects…I also recommend to combine them to one figure and distinguish between A and B (e.g. Figure 2A à Site A and Figure 2B à Site B), because the information is identical in both.

A: *Thank you for your comment! We have followed your suggestion!*

Q: Figure 4:

please check upper and lower case after colon.

A: *Thank you for your comment! We have corrected it.*

Q: Method, Line 232

"At each cross-section had 4-5 samples were taken" à remove "had"

A: *Thank you for your comment! We have corrected it.*

Q: Methods, Line 286-287

I recommend not to use the word "approximately" here but use the correct percentages for training and validation. Otherwise, the reader gets the impressions that the authors are not sure about the percentage allocation.

A: *Thank you for your comment! We have corrected it. Even though the exact number of these images (46) was published, but we could not find information*

Q: Method, Line 289-292

What are these hyperparameters and what do they change, why do they to be changed? What are the numbers related learning rate 0.01, what is a decay of 0.1 and what a batch size of 15. What is the cross-entropy loss function doing? Well, I am not an expert of AI or any other kind of learning algorithms but a few words explaining the numbers and expressions would significantly increase the comprehensiveness.

A: *Thank you for your comment! We have updated the manuscript and added some explanation to the hyperparameters. Line 326 – 338.*

Q: Results and Discussion, Line 308, 316

Remove the "." After "Figure 5"… actually, sometimes it is written Fig. and sometimes Figure, please check the entire manuscript for consistency (and if necessary author guidelines).

A: *Thank you for your comment! We have corrected and updated the manuscript accordingly.*

Q: Results and Discussion, Line 310

Why is the expression "pixel accuracy" written in italic?

A: *Thank you for your comment! It was a mistake. We have corrected and updated the manuscript accordingly.*

Q. Figure 5

The caption should describe the 4 single columns. Not necessary to include an evaluation into the caption.

A: *Thank you for your comment! We have corrected and updated the manuscript accordingly.*

Q: Figure 7, 11

Maybe better: "The fractions of the physical samples are shown as verticals". I recommend to increase the thickness of the verticals for better visualisation.

A: *Thank you for your comment! We have corrected and updated the manuscript accordingly.*

Q: Results and Discussion, Line 358

"The images of the bed from the sampling points are in shown in Figure 9" à Remove the first "in"

A: *Thank you for your comment! We have corrected and updated the manuscript accordingly.*

Q: Results and Discussion, Line 367

please correct "thalweq"

A: *Thank you for your comment! We have corrected and updated the manuscript accordingly.*

Q: Results and Discussion, Line 400-401

please correct "of the of the sediment"

A: *Thank you for your comment! We have corrected and updated the manuscript accordingly.*

Q: Results and Discussion, Line 406-407

I guess the authors mean that the weighting of the physical samples resulted in a mass distribution!?

A: *Thank you for your comment! The manuscript mentioned it as: "The collected samples were analysed in laboratory by drying, sieving, and weighing to provide local grain size distribution." in ch. 3.2.*

*To avoid confusion, we have removed "volumetric distribution" and replaced it with "weight distribution". Line 505 and Line 636.*

Q: Results and Discussion, Line 430-432

In this sentence, the authors describe the results of the other sections in the appendix and find a "well-captured" performance of the AI. What is the base for a well-captured AI? Why are not the values mentioned in the "Method"-Section published?

A: *Thank you for your comment! We have introduced Fig. 22 and Fig 23. Explanation and reasoning for these new figures are also given: Line 21 – 24 (Abstract), Line 552 – 575; Line 642 – 645.*

Q: Results and Discussion, Line 439-442

This statement need to be proven by data. How many successful detection did the AI (percentage)…what means successful in this way? I strongly recommend that the authors underpin such statements with data.

A: *Thank you for your comment! We have introduced Fig. 22 and Fig 23. Explanation and reasoning for these new figures are also given: Line 21 – 24 (Abstract), Line 552 – 575; Line 642 – 645.*

Q: Results and Discussion, Line 454-456

Did the authors investigated the effect of lighting? E.g. experiments in shallow clear rivers without lighting compared to the dark deeper investigations? What are the limitations for the AI regarding the lighting beam? Would it be feasible that the AI learn what grey colour value lighting is? It would be interesting to get some more details on this aspect! Especially what can be done to avoid it…

A: *Thank you for your comment! As of now, only plans exists for further investigating the question of lighting. Unfortunately, the scope of the present manuscript does not include further analysis for this. As we mention in the manuscript, we think one practical solution would be to use lamps with uniform light, ones which are less focused. Other option could be to teach the AI to recognise the light beam and torching effect and simply remove the detected area from calculation. This of course would decrease the effective analysed area and the representativeness.*

Q: Results and Discussion, Line 482-483

I do not agree that the mathematical values are senseless, although they are of course not a perfect tool for validation purposes. However, they allow for analyses in a certain value range (e.g. 0.4 – 0.6, as mentioned earlier by the authors) and to investigate all data and plot them in a meaningful way (e.g. box plots). This would be highly beneficial…

A: *Thank you for your comment! We have introduced Fig. 22 and Fig 23. Explanation and reasoning for these new figures are also given: Line 21 – 24 (Abstract), Line 552 – 575; Line 642 – 645.*

Q: Results and Discussion, Line 542ff

Structure-from-motion required overlapping images from different perspectives, but the camera provides images of the riverbed from only one side, I cannot not imagine that this can be used to generate accurate enough 3D-models of the riverbed.

A: *Thank you for your comment! As the camera passes over an object (sediment particle) in its path, several consecutive images will be taken of the object along the direction of the camera movement. Of course, it means higher uncertainty in the transverse direction. In this sense however, it is the same as airborne Lidar surveys where the drones are also not sweeping through the area from many perspectives, but rather carry out single-flight measurements. For estimating bed roughness for example, it is enough to have a firm grasp in the direction of the camera movement. Partially for this reason, we chose the directional roughness calculation method in our earlier study (Ermilov et al., 2020.) where we tested underwater Sfm. If one wishes to have better 3D results in the transverse, then applying two cameras simultaneously, with overlapping FOV would be beneficial. This is part our plans,*

*partially for this very reason, but also to increase the size of the analysed area and improve the representativity of the method.*

Citation: https://doi.org/10.5194/esurf-2022-56-RC2

---

## Referee Report (RR1)

**Review of Automated riverbed material analysis using Deep Learning on underwater images – Ermilov et al.**

This paper presents a study using artificial intelligence to obtain the composition of underwater riverbed materials. The method is able to capture both sand, gravel and comble fractions relatively successfully when compared with physical samples and photos analysed using a previously published wavelet-based approach. The algorithm used videos taken underwear along the Hungarian Danube to both train and validate the technique. This would allow for large sections of riverbed, which cannot be measured using standard techniques, e.g. underwater, large areas and often without scale, to be analysed. This research has built on previous foundations detecting grain size automatically and recent developments in AI to provide an indication of riverbed composition on a larger scale than previously attainable.

Thank you for this interesting and novel paper, which utilises modern methods to advance the field of fluvial geomorphology. It is evident that the authors have spent a significant amount of time developing the methodology, collecting the training and validation datasets, and analysing the results, which appear both exciting and promising. I think this paper will be of interest to the readers of this journal. However, I have some general suggestions listed below which I believe will improve the quality and readability of the manuscript.

- The use of figures within the manuscript is ineffective at times. In my opinion the manuscript has too many figures (20+) which detracts from the important figures in the manuscript and reduces readability of the paper. In the line-by-line comments I have provided suggestions on how to combine existing figures or when to make use of the supplemental.
- The terminology used to describe the sample sites is unclear and should be changed so that it is more intuitive for the reader. A simple format (such as AII-1, AII-2 and BII-1, BII-2) would help readers to compare between original photos and points along the channel. The current terms (e.g. VM4, 3/4) could not be easily referred to by someone who was not involved in the study.
- In general, the manuscript was easy to follow and well-written. However, certain sections could benefit from further attention.
    - I would personally use the figures as references as opposed to the initial subject of the sentence as this can break up the flow of the manuscript, e.g., "Figure 9 presents…" would change to "We applied a moving-averaged based smoothing technique to obtain… (Fig. 9)." This is already in parts of the manuscript, but I think more consistent use would help improve the manuscript.
    - The Results and Discussion section (Section 4.2 currently) could be improved by not separating the results by study location and instead integrating all three locations (including those in the appendix) into a discussion about how the AI algorithm compared to the different methods. The results and discussion could then be titled *3*.2. Comparison of methods with subheadings for each method type (e.g. 3.2.1 Physical samples, 3.2.2 Wavelet analysis). The authors could then compare and contrast the different methods for each location to avoid repetition. Figures 22 and 23 could also be integrated into the respective sections.
- The method appears to be successful by current standards however I had some trouble understanding exactly how the accuracy scores were calculated and what was being shown in Figures 22 and 23 (though I really like these figures and think they make a great addition to the manuscript following the initial revision). For example, why are there only 11 points when many more photos were used for validation (>2000)? This should be clarified to

ensure that readers understand the accuracy when using the method. The manuscript may benefit from a plot demonstrating the accuracy for all tested images (e.g. gravel content vs IoU) as this may show where the algorithm performs well and less well.

**Line by line comments**

Title suggestion: Automated riverbed *composition* (?) analysis using Deep Learning on underwater images

Line 10: Remove indeed

Line 12: Add by after "…overcome this issue…"

Line 22 and 23: After reading the manuscript I am unsure what these percentages are based on, is it 64% of 11 photos? It would be great to add an n value here to show what proportion of the dataset as been used here.

**Introduction**

Lines 31 to 40: Whilst the content in this paragraph was well placed and relevant for the manuscript, I think the paragraph could be written more clearly. For example, without the use of etc. and with references after the examples (e.g. sentence on line 37 and 38 did not give reference but gave lots of examples).

Line 38: It would be helpful to break down sediment composition early in the manuscript, for example "sediment composition (sand and gravel content)" to make clear what you are measuring.

Line 59: I would use "Section" as opposed to "Chapter" in a manuscript for this journal, but a reference at the end of the following sentence would be sufficient here. Section 2 has also been removed following the previous review stage so the reference to that section no longer fits here.

Line 62: Thank you for breaking down the previous research on alternative methodologies for measuring grain size. A table with a breakdown of the previous research would be very helpful for readers of the manuscript who are not familiar with the past work. The table could include the limitations outlined in bullet points (lines 89 to 100).

Lines 69 to 71: Sorry this is not my area of expertise, so I found the sentence starting "Researchers found that…" unclear. Is there a specific article which "researchers" is used to refer to? And what are the specific coefficients used to obtain, i.e. why convert the signal strength?

Line 75: Why could gravel not be distinguished strongly from sand?

Line 80: picture to image?

Line 83: "Both" as opposed to "The above-mentioned"

Line 87: If a table is created as suggested in the Line 62 comment above, I think a reference to the table here would work well and then the bullet points could be reduced to a sentence/found in the table.

Lines 152 to 161: I would specifically state that that this method will be used to obtain pebble and sand fraction as that is really exciting and should not be understated.

Line 163: earlier is repeated in the sentence, remove one.

Line 172: The reference to the third chapter is no longer valid as the second section (originally literature review has been combined with the introduction). I would also use Section here not Chapter. I don't think this paragraph is necessary for the readability of the manuscript but am happy to leave to the authors discretion.

**Methods**

Line 179: All the numbering from this point is incorrect, methods should be 2. Please check this.

Line 182: "one" seems vague, does this mean video? Or dataset? Or collection of videos/transects?

Line 182: "the second one", can remove "one" here, once the first "one" is replaced with a more specific term.

Figure 1: I like how the Danube has been emphasised.

Line 191: Change "similarly to" to "and".

Line 200: I would move $SSC_{survey}$ to in brackets following the first mention of suspended sediment concentrations in the previous line (199).

Figure 2: Great figure, really clear. I would add A and B to the different site maps so that each site can be referred to in the text.

Line 284: Videos might work better here as opposed to footages?

Page 11, Second Paragraph: Line numbers do not work here, potentially due to figure placement.

In the sentence "These steps were followed by the annotation, where we distinguished ten classes", the ten classes should be written as a list. Alternatively, they could be included in Figure 4, in which case the authors could refer to the figure.

A space is also needed between "the" and "4" further in the paragraph in the sentence "In total, a…".

Figure 4: As someone not familiar with Deep Learning algorithms, I appreciated the step-by-step flow chart. I had a few suggestions that might help readers unfamiliar to better understand the process.

Data creation box:

- Change "From underwater videos" to "Uses underwater videos".
- Change "fix" to "fixed"

White balance upgrade

- I found the description in the first bullet point unclear and wondered if the step could be clarified.

Data annotation

- You could list the ten classes here if not in the text.

Visualization and analysis

- I think the overall accuracy is also referred to in the text, however I think more information should be given in the box about how this is calculated. Is this the accuracy as in number of

gravel, cobble and sand pixels in each image when manually mapped and when using AI? Please clarify.

Line 353: Remove "by the authors of present manuscript".

**Results and discussion**

Line 355: 4 to 3.

Line 357: I think this paragraph could be rewritten in parts. I was unclear about whether the datasets being discussed were the training dataset or the validation photos. The first sentence in the paragraph could be used to clarify this by providing a simple, general statement about how successful the method was at recording the grain size fractions in the training/validation dataset.

Line 358: Is the validation set the grain size fractions obtained using the wavelet method? Please clarify.

Line 360: "over-all" to "overall.

Line 361: Is this the percent of pixels correctly identified or the number of photos where the method was successful? Please clarify. A plot showing the variation in the accuracy value and IoU value per photo would be interesting and may more clearly present areas where the algorithm is more and less (e.g. poorer quality images, high SSC?) accurate.

Line 361: Please add the n value for the 96% accuracy, e.g. is n = 2957 – the number of photos? Or is it the number of photos used for training?

Figure 5: To avoid a lot of figures in the manuscript, which dilute the message, I would suggest combining Figures 5 and 6 and using A, B, C, D and E to separate each image row.

Line 387: Comparison as opposed to intercomparison. Please see general comments for a suggestion on how to change this section of the manuscript.

Line 388: I really like this paragraph. Perhaps this would be a good opportunity to introduce Figures 22 and 23.

Line 404: Is the 15 m smoothing transferable across sites/equipment or is there a specific reason for its use here?

Line 405: "are the ones being compared" to "are used to compare…"

Figure 7: Combine Figures 7 and 9. Both are really useful, and I think combining the plots as a two-panel figure would be more effective and allow for each reference between the sampling location and the sediment composition measured. The plots could also be on the same scale so that comparison is simple. As a side note, Figure 9 is really effective.

Figure 8: Whilst this is an interesting figure, particularly as this is a methods-based paper, I think it would be better placed in the supplemental, which is an option in ESurf and referred to in the text as "Figure SX in the Supplement".

Line 429: Change "an" to "and"

Line 430: Change "eye" to "visual"

Line 438: Change "with" to "by".

Line 446 and Figure 11: Move to supplement.

Line 447 to 448: Add "we broke the surface amour to showcase the presence of the underlying finer fractions" to the figure caption and remove from the main text.

Figures 13 and 14: Can be combined and moved the supplemental.

Line 481: Remove "again"

Figure 15 and 16: Combine but keep within main text, see comments on Figure 7 and naming of samples at the start of this review.

Line 510: Use 19b as opposed to 19/b when referring to figure.

Figure 17: Move to supplemental.

Figure 20: These figures could be combined with the earlier Figure 5 (and Figure 6, see earlier comment) as two additional rows to the plot and then referred to using letters.

Figure 21: Move to supplement.

Line 539: In this journal, I think the "Appendix" is considered the "Supplemental" so this should be changed.

Line 550: Is "efficiency" the correct term here? Sorry I am not familiar with vocabulary used to describe Deep Learning algorithms. "Applicability" came to mind as the sentence appeared to be detailing which photos could be used with this method, but "efficiency" may work within the context.

Line 553: Change "belonged" to "belonging".

Line 555: What does the term points refer to in this paraph, it is unclear, photos? Study locations? Why are only 11 included if so? And how many 'points' were there in total? Add n = xx. Use of a clear sample naming procedure would also improve clarity in this paragraph.

Figure 22: I really like this plot and think it is a good addition to the paper. However, I find it difficult to fully understand what is being shown. Are the plots showing the relative proportions of each size fraction based on the 11 points (photos or sites or both?) when using the AI detection in this paper and physically sampling from the riverbed? I would advise splitting this figure into three figures (A, B and C) and using the shape/colour of the points to represent the different samples as it is unclear what has been plotted. The same could also be done for Figure 23. I think the figure caption could include more detail to help guide the readers and the axes titles could also be clearer. For example, if three figures are used the author could plot "Physical samples gravel fraction (%)".

Also why only select the most comparable images? Is there benefit to sharing the full dataset? Maybe you could include samples with bed armouring and thin sand layers in a grey colour/different shape so readers can see the impact of sampling in these locations.

Line 567: Similarly, the use of points needs to be clarified earlier or changed.

Line 567: Change "while neither…" to "unlike the AI and physical samples". And remove "did so".

Figure 23: I think three figures (as explained for Figure 22) would also be interesting here and show if there are any differences across study sites. The x axis also states physical but the caption says wavelet analysis, please check this.

Line 596: Move this statement to the methods section as well as kept here where it is being discussed.

Line 622: remote "i.e.".

Line 625: The sentence starting "However, the latter…" is quite vague and I think it would be good to clarify.

Line 640: Remove "," after size.

Line 642: Echoing the comments made on the abstract, is this just the 11 points? Are these transects? I think an n value for this statistic relative to the total number of photos tested would be useful.

Line 643: typo "nut" to "not".

Line 643: "surpassed" to "surpass".

Line 665: Chapter to Section

---

## Author Response (AR2)

Review of Automated riverbed material analysis using Deep Learning on underwater images –

Ermilov et al.

**General**

R: The use of figures within the manuscript is ineffective at times. In my opinion the manuscript has too many figures (20+) which detracts from the important figures in the manuscript and reduces readability of the paper. In the line-by-line comments I have provided suggestions on how to combine existing figures or when to make use of the supplemental.

A: Thank you very much! I agreed with your insights and tried my best to follow your instructions in general!

R: The terminology used to describe the sample sites is unclear and should be changed so that it is more intuitive for the reader. A simple format (such as AII-1, AII-2 and BII-1, BII-2) would help readers to compare between original photos and points along the channel. The current terms (e.g. VM4, 3/4) could not be easily referred to by someone who was not involved in the study.

A: Thank you very much! I have changed the names in both the text and on the figures.

R: The Results and Discussion section (Section 4.2 currently) could be improved by not separating the results by study location and instead integrating all three locations (including those in the appendix) into a discussion about how the AI algorithm compared to the different methods. The results and discussion could then be titled 3.2. Comparison of methods with subheadings for each method type (e.g. 3.2.1 Physical samples, 3.2.2 Wavelet analysis). The authors could then compare and contrast the different methods for each location to avoid repetition. Figures 22 and 23 could also be integrated into the respective sections.

A: Thank you very much! I have followed your instructions and rearranged Section 3.2 entirely.

R: The method appears to be successful by current standards however I had some trouble understanding exactly how the accuracy scores were calculated and what was being shown in Figures 22 and 23 (though I really like these figures and think they make a great addition to the manuscript following the initial revision). For example, why are there only 11 points when many more photos were used for validation (>2000)? This should be clarified to ensure that readers understand the accuracy when using the method. The manuscript may benefit from a plot demonstrating the accuracy for all tested images (e.g. gravel content vs IoU) as this may show where the algorithm performs well and less well.

A: Thank you for your comment! I have added more explanation to the text (e.g. Line 16 – 17; Line 499 – 510 and Table 2).

**Line by line comments**

R: Title suggestion: Automated riverbed composition (?) analysis using Deep Learning on underwater images

A: Thank you very much! I have changed it accordingly.

R: Line 10: Remove indeed

A: Thank you very much! I have changed it accordingly.

R: Line 12: Add by after "…overcome this issue…"

A: Thank you very much! I have changed it accordingly.

R: Line 22 and 23: After reading the manuscript I am unsure what these percentages are based on, is it

A: Thank you very much! I have changed it accordingly.

R: 64% of 11 photos? It would be great to add an n value here to show what proportion of the dataset as been used here.

A: Thank you very much! I have changed the manuscript accordingly. I have added Table 2, for example. Furthermore, I tried my best to further ellaborate on it in the text. Line 16-17, Line 23-26, Line 485, Line 496-497, Line 508- 512

**Introduction**

R: Lines 31 to 40: Whilst the content in this paragraph was well placed and relevant for the manuscript, I think the paragraph could be written more clearly. For example, without the use of etc. and with references after the examples (e.g. sentence on line 37 and 38 did not give reference but gave lots of examples).

A: Thank you very much! I agreed with your insights and tried my best to follow your instructions! Line 41-43

R: Line 38: It would be helpful to break down sediment composition early in the manuscript, for example "sediment composition (sand and gravel content)" to make clear what you are measuring.

A: Thank you very much! I have changed it accordingly. Line 44

R: Line 59: I would use "Section" as opposed to "Chapter" in a manuscript for this journal, but a reference at the end of the following sentence would be sufficient here. Section 2 has also been removed following the previous review stage so the reference to that section no longer fits here.

A: Thank you very much! I have changed the manuscript accordingly (from Chapter to Section everywhere, also adjusted the Section numbers).

R: Lines 69 to 71: Sorry this is not my area of expertise, so I found the sentence starting "Researchers found that…" unclear. Is there a specific article which "researchers" is used to refer to? And what are the specific coefficients used to obtain, i.e. why convert the signal strength?

A: Thank you very much! I have changed the manuscript accordingly. Line 76 - 99

R: Line 75: Why could gravel not be distinguished strongly from sand?

A: Thank you very much! I have changed the manuscript accordingly. Line 84 – 86

R: Line 80: picture to image? Line 83: "Both" as opposed to "The above-mentioned"

A: Thank you very much! I have changed the manuscript accordingly.

R: Lines 152 to 161: I would specifically state that that this method will be used to obtain pebble and sand fraction as that is really exciting and should not be understated.

A: Thank you very much! I have changed the manuscript accordingly. Line 178 – 179

R: Line 163: earlier is repeated in the sentence, remove one. Line 172: The reference to the third chapter is no longer valid as the second section (originally literature review has been combined with the introduction). I would also use Section here not Chapter. I don't think this paragraph is necessary for the readability of the manuscript but am happy to leave to the authors discretion.

A: Thank you very much! I have changed the manuscript accordingly. I have decided to leave the paragraph in since if I remember correctly it was asked by the reviewers in the Public Discussion phase. I have corrected the reference and removed the repeated word.

**Methods**

R: Line 179: All the numbering from this point is incorrect, methods should be 2. Please check this.

A: Thank you very much! I have changed the manuscript accordingly.

R: Line 182: "one" seems vague, does this mean video? Or dataset? Or collection of videos/transects? Line 182: "the second one", can remove "one" here, once the first "one" is replaced with a more specific term.

A: Thank you very much! I have changed the manuscript accordingly. Line 197 – 200

R: Figure 1: I like how the Danube has been emphasised.

A: Thank you for your feedback!

R: Line 191: Change "similarly to" to "and". Line 200: I would move $SSC_{survey}$ to in brackets following the first mention of suspended sediment concentrations in the previous line (199). Figure 2: Great figure, really clear. I would add A and B to the different site maps so that each site can be referred to in the text. Line 284: Videos might work better here as opposed to footages?

A: Thank you very much! I have replaced the mentioned words, added SSC to the brackets (Line 215) and updated Figure 2 accordingly.

==**Attention!** Seems like Table 1 was divided between two pages when I saved my final version of the PDF. Terribly sorry for that! I will correct it in the next review phase!==

R: Page 11, Second Paragraph: Line numbers do not work here, potentially due to figure placement.

A: Thank you for your feedback! Yes, it happens because of the long figure.

R: In the sentence "These steps were followed by the annotation, where we distinguished ten classes", the ten classes should be written as a list. Alternatively, they could be included in Figure 4, in which case the authors could refer to the figure. A space is also needed between "the" and "4" further in the paragraph in the sentence "In total, a…".

A: Thank you very much! I have changed the manuscript accordingly. I have updated Figure 4 with the classes. However, I insisted on mentioning them in the text as well, if that is not a problem I would keep it there too!

R: Figure 4: As someone not familiar with Deep Learning algorithms, I appreciated the step-by-step

flow chart. I had a few suggestions that might help readers unfamiliar to better understand the

process.

Data creation box:

- Change "From underwater videos" to "Uses underwater videos".

- Change "fix" to "fixed"

White balance upgrade

- I found the description in the first bullet point unclear and wondered if the step could be

clarified.

Data annotation

- You could list the ten classes here if not in the text.

Visualization and analysis

- I think the overall accuracy is also referred to in the text, however I think more information should be given in the box about how this is calculated. Is this the accuracy as in number of gravel, cobble and sand pixels in each image when manually mapped and when using AI? Please clarify.

A: Thank you very much! I have changed the manuscript accordingly. I have updated Figure 4.

R: Line 353: Remove "by the authors of present manuscript".

A: Thank you very much! I have changed the manuscript accordingly.

**Results and discussion**

R: Line 355: 4 to 3.

A: Thank you very much! I have changed the manuscript accordingly.

R: Line 357: I think this paragraph could be rewritten in parts. I was unclear about whether the datasets being discussed were the training dataset or the validation photos. The first sentence in the paragraph could be used to clarify this by providing a simple, general statement about how successful the method was at recording the grain size fractions in the training/validation dataset.

A: Thank you very much! I have changed the manuscript accordingly. Line 378 - 385

R: Line 358: Is the validation set the grain size fractions obtained using the wavelet method? Please clarify.

A: Thank you very much! I have changed the manuscript accordingly. It is better to think of DL validation as a self-validation. It resembles a student who learnt something at home (training process, with 80% of the 14,784 human-annotated images prepared by the teacher (us). During validation, the student writes a test and receives 2957 raw images (20% of the 14,784 images) and has to classify them based on what he/she learnt. The teacher (us) of course has the answers (this 2957 images were annotated during annotation) and can compare the student's answers to this solution-sheet. Relevant parts of the manuscript: Figure 4 – Training box; Line 342 – 344, Line 378 – 382

R: Line 360: "over-all" to "overall.

A: Thank you very much! I have changed the manuscript accordingly.

R: Line 361: Please add the n value for the 96% accuracy, e.g. is n = 2957 – the number of photos? Or is it the number of photos used for training?

A: Thank you very much! I have changed the manuscript accordingly. Line 382 – 385

R: Figure 5: To avoid a lot of figures in the manuscript, which dilute the message, I would suggest combining Figures 5 and 6 and using A, B, C, D and E to separate each image row.

A: Thank you very much! I have changed the manuscript accordingly.

R: Line 387: Comparison as opposed to intercomparison. Please see general comments for a suggestion on how to change this section of the manuscript.

A: Thank you very much! I have changed the manuscript accordingly.

R: Line 388: I really like this paragraph. Perhaps this would be a good opportunity to introduce Figures 22 and 23.

A: Thank you very much! I have changed the manuscript accordingly. Figure 9 and Figure 13 has been introduced in this Section, I also rearranged the whole Section.

**Attention!** Seems like some of the figure-caption style was not automatised when I changed the manuscript and it did not get my attention (Figure 9, Figure 12. Again, I'm sorry for that… I will correct them in the next review phase.

R: Line 404: Is the 15 m smoothing transferable across sites/equipment or is there a specific reason for its use here?

A: Thank you very much! I have changed the manuscript accordingly. Line 428 – 433

R: Line 405: "are the ones being compared" to "are used to compare…"

A: Thank you very much! I have changed the manuscript accordingly.

R: Figure 7: Combine Figures 7 and 9. Both are really useful, and I think combining the plots as a two-panel figure would be more effective and allow for each reference between the sampling location and the sediment composition measured. The plots could also be on the same scale so that comparison is simple. As a side note, Figure 9 is really effective.

A: Thank you very much! I have changed the manuscript accordingly.

R: Figure 8: Whilst this is an interesting figure, particularly as this is a methods-based paper, I think it would be better placed in the supplemental, which is an option in ESurf and referred to in the text as "Figure SX in the Supplement".

A: Thank you very much! I have changed the manuscript accordingly.

R: Line 429: Change "an" to "and"

Line 430: Change "eye" to "visual"

Line 438: Change "with" to "by".

A: Thank you very much! I have changed the manuscript accordingly.

R: Line 446 and Figure 11: Move to supplement.

A: Thank you very much! I have changed the manuscript accordingly.

R: Line 447 to 448: Add "we broke the surface amour to showcase the presence of the underlying finer fractions" to the figure caption and remove from the main text.

A: Thank you very much! I have changed the manuscript accordingly.

R: Figures 13 and 14: Can be combined and moved the supplemental. Line 481: Remove "again". Figure 15 and 16: Combine but keep within main text, see comments on Figure 7 and naming of samples at the start of this review. Line 510: Use 19b as opposed to 19/b when referring to figure. Figure 17: Move to supplemental. Figure 20: These figures could be combined with the earlier. Figure

5 (and Figure 6, see earlier comment) as two additional rows to the plot and then referred to using letters. Figure 21: Move to supplement.

A: Thank you very much! I have changed the manuscript accordingly.

R: Line 539: In this journal, I think the "Appendix" is considered the "Supplemental" so this should be

changed.

A: Thank you very much! I checked the Formatting Requirements on ESurf and there they ask it to be Appendix. ([https://www.earth-surface-dynamics.net/submission.html#templates](https://www.earth-surface-dynamics.net/submission.html#templates)) at Manuscript composition: 7. Appendices

R: Line 550: Is "efficiency" the correct term here? Sorry I am not familiar with vocabulary used to describe Deep Learning algorithms. "Applicability" came to mind as the sentence appeared to be detailing which photos could be used with this method, but "efficiency" may work within the context. Line 553: Change "belonged" to "belonging".

A: Thank you very much! I have changed the manuscript accordingly.

R: Line 555: What does the term points refer to in this paraph, it is unclear, photos? Study locations?

Why are only 11 included if so? And how many 'points' were there in total? Add n = xx. Use of a clear

sample naming procedure would also improve clarity in this paragraph.

A: Thank you very much! I have changed the manuscript accordingly. Line 484 – 507

R: Figure 22: I really like this plot and think it is a good addition to the paper. However, I find it difficult to fully understand what is being shown. Are the plots showing the relative proportions of each size fraction based on the 11 points (photos or sites or both?) when using the AI detection in this paper and physically sampling from the riverbed? I would advise splitting this figure into three figures (A, B and C) and using the shape/colour of the points to represent the different samples as it is unclear what has been plotted. The same could also be done for Figure 23. I think the figure caption could include more detail to help guide the readers and the axes titles could also be clearer. For example, if three figures are used the author could plot "Physical samples gravel fraction (%)". Also why only select the most comparable images? Is there benefit to sharing the full dataset? Maybe you could include samples with bed armouring and thin sand layers in a grey colour/different shape so readers can see the impact of sampling in these locations. Line 567: Similarly, the use of points needs to be clarified earlier or changed.

A: Thank you very much! I have changed the manuscript accordingly. I have improved Figure 9 and Figure 13 by addind the name of sampling points next tot he data points where they are taken from. I also extended their figure captions. I also further ellaborated on it in the text: Line 484 – 507, Line 514 – 517, Line 551 – 557

R: Line 567: Change "while neither…" to "unlike the AI and physical samples". And remove "did so".

A: Thank you very much! I have changed the manuscript accordingly.

R: Figure 23: I think three figures (as explained for Figure 22) would also be interesting here and show if there are any differences across study sites. The x axis also states physical but the caption says wavelet analysis, please check this.

A: Thank you very much! Similarily to Figure 9, I wished to help the case by including the name of the sampling points at their respective data points.

R: Line 596: Move this statement to the methods section as well as kept here where it is being

discussed. Line 622: remote "i.e.". Line 625: The sentence starting "However, the latter…" is quite vague and I think it would be good to clarify. Line 640: Remove "," after size.

A: Thank you very much! I have changed the manuscript accordingly.

R: Line 642: Echoing the comments made on the abstract, is this just the 11 points? Are these transects? I think an n value for this statistic relative to the total number of photos tested would be useful.

A: Thank you very much! I have changed the manuscript accordingly. Please, see my earlier responses. Line 484 – 507, Line 514 – 517, Line 551 – 557 and Figure 9, Figure 13.

R: Line 643: typo "nut" to "not".

Line 643: "surpassed" to "surpass".

Line 665: Chapter to Section

A: Thank you very much! I have changed the manuscript accordingly.

R: Line 62: Thank you for breaking down the previous research on alternative methodologies for measuring grain size. A table with a breakdown of the previous research would be very helpful for readers of the manuscript who are not familiar with the past work. The table could include the limitations outlined in bullet points (lines 89 to 100). Line 87: If a table is created as suggested in the Line 62 comment above, I think a reference to the table here would work well and then the bullet points could be reduced to a sentence/found in the table.

A: Thank you very much! I am sorry, I was not sure what you wished to see here exactly. Can I ask for further clarification? The bulletpoints of limitations only refer to the image-based alternative

approaches. Hence the Table should include these studies in my opinion only. Should it be a 2-row table, where first row is the Referred study, and the second is its limitation? Or maybe something else? Thank you in advance for the further clarification!

R: Line 361: Is this the percent of pixels correctly identified or the number of photos where the method was successful? Please clarify. A plot showing the variation in the accuracy value and IoU value per photo would be interesting and may more clearly present areas where the algorithm is more and less (e.g. poorer quality images, high SSC?) accurate.

A: Thank you very much! I have changed the manuscript accordingly. Line 378 – 388. Regarding the preparation of this kind of figure I was not able to compile:

Unfortunately the selection of the validation set (2957 images) is randomized, and when I prepared this I did not think of preparing a figure like this. It further adds tot he difficulties that they are not even whole images but augmented ones (see Augmentation) and I had trouble with identifying each of these 2957 images point locations in time for the submission. On the other hand, these points (due to random selection) would not clearly show areas, but would rather act as a point cloud between the cross-sections of Site C and proportion of Site A. Maybe some of them even belonging to the same image. I feel doubtful if it would show some sort of correlation with the SSC, because it also includes the effect of the camera getting too far away (bed forms, channel shape) and the Deep Learning algoritm's inefficiency/efficiency. Anyhow, if required, I will definitely try my best and continue the preparation of this figure for the next review phase.

---

## Author Response (AR3)

Minor comments

Q: Please carefully reread and edit the results and discussion. There are many wordings which indicate personal opinions and instances where results need to be presented in a more scientific way. For example, line 673f "Of course, as a trade-off, the method, as of now, cannot reconstruct detailed grainsize distribution, ...". Please try to be concise here, avoid repetitions, and quantitatively underpin your statements. Also, several fill-words such as "It should be noted" (L669) should be avoided as well as subjective expressions such as "great alternative" (L661).

A: Thank you for your comment! The chapter has been reviewed and I changed the wording in several places.

Q: In Table 1, you refer to Q1% as the discharge with 1% probability. I assume that this is rather the 1-in-a-100 year flood, or the flood with a 1% annual exceedance probability (or 1% AEP flood).

A: Thank you for your comment! Yes, you are correct. I have corrected the expression accordingly.

Q: In Figure 2, please also indicate the coordinate reference system. From the coordinates, I can see that it is not UTM coordinates.

A: Thank you for your comment! I have indicated the reference system accordingly.

Q: Figure 4 might work better with a horizontal layout. But this might be handled during production.

A: Thank you for your comment! I agree with your idea, however I am not sure how the page setup is going to be edited before printing, so as of now I did not change Figure 4, leaving itt o the production stage.

Q: Figure 7 and 8 (and figures in the Appendix) might be difficult to read for people with red-green color vision deficiency. Please consider to use different colors as also notified by the editorial team during review file validation.

A: Thank you for your comment! I have added distinguishable symbols and hatches to the color-scheme, hence I decreased the importance of the colors themselves. The updated figure are: 7., 8., A1, C1, C2, D1, D2, E1, E2,

Q: Code and data availability: Make sure to use permanent repositories for the code and data such as zenodo or figshare

A: Thank you for your comment! I have uploaded the files to Figshare and updated the code and data availability links.

---

## Author Response (AR4)

Minor comments

**1) Specific instructions of the reviewer**

Q: Line 9: high instead of great

A: Thank you for your comment! We have followed your suggestion.

Q: Line 10f: Conventional sampling methods are inadequate and time-consuming for effectively capturing the variability of bed surface texture in these situations.

A: Thank you for your comment! We have followed your suggestion.

Q: Line 12: In this study, we overcome this issue by adopting an image-based, Deep Learning (DL) algorithm (delete the rest of this sentence)

A: Thank you for your comment! We have followed your suggestion.

Q: Line 13: You introduce an abbreviation for Deep Learning (DL) in line 13 and 129. Please use this abbreviation from here on.

A: Thank you for your comment! We have followed your suggestion.

Q: Line 15: that were taken along cross-sections underwater in the Danube river. 27 river bed samples were collected and analyzed for validation.

A: Thank you for your comment! We have followed your suggestion.

Q: Line 18: how dense is "very dense". Please try to make quantitative rather than qualitative statements.

A: Thank you for your comment! We have followed your suggestion.

Q: Line 23: Delete sentences: This meant a total of 27 ...

A: Thank you for your comment! We have followed your suggestion.

Q: Line 24: I wouldn't call this data outlier. Perhaps rephrase this part:

After correcting for samples affected by bed armouring, comparison of the DL approach with 14 physical samples yield a mean classification error of 4.5%.

A: Thank you for your comment! We have followed your suggestion.

Q: Line 27: From here on, there is a lot detail for the abstract.

A: Thank you for your comment! I have removed the „Furthermore, comparison with the wavelet-based image processing justified the selection of the outlier points earlier, as its results matched closely with the DL detections in these purely gravel-covered points and showed no sign of finer fractions, univocally opposing the content of the physical samples." Line 27 – 30.

Q: Line 64: remove intensively

A: Thank you for your comment! We have followed your suggestion.

Q: Line 65: remove Major

A: Thank you for your comment! We have followed your suggestion.

Q: Line 175: The goal of this study is to ... and attempt to solve the shortcomings of previous ...

A: Thank you for your comment! We have followed your suggestion.

Q: Line 187: study instead of manuscript

A: Thank you for your comment! We have followed your suggestion.

Q: Line 190: varies instead could vary

A: Thank you for your comment! We have followed your suggestion.

Q: Line 190: in most previous studies

A: Thank you for your comment! We have followed your suggestion.

Q: Line 243: remove: Of course

A: Thank you for your comment! We have followed your suggestion.

Q:  Line 619: four approaches were adopted:

A: Thank you for your comment! We have followed your suggestion.

Q:  Line 669-670: Provide references for this statement.

Thank you for your comment! This section is referencing Section 1., where several previous studies have been introduced already. We now modified the sentence to make it clearer: *As we discussed in Section 1., earlier DL methods for sediment analysis (e.g., Soloy et al., 2020) all applied fixed camera heights and/or provided scaling for the AI.*

Q: Line 672-675: Delete the sentence: Hence, it is of major importance that this manuscript introduces ... Instead write: By avoiding the need for a scale, our method is faster and simpler to use. As a drawback, our method does not reconstruct ...

Q: Line 681: discriminated instead of adapted?

A: Thank you for your comment! We have followed your suggestion.

Q: Line 689: imaging techniques

A: Thank you for your comment! We have followed your suggestion.

Q: Line 692: LS-PIV? This abbreviation remains unclear.

A: Thank you for your comment! We have followed your suggestion.

Q: Line 703: please check, whether representativeness is the right word here.

Thank you for your comment! Both representativeness (Chasalow & Levy, 2021) and representativity (Booth & Gerland, 2015) are correct/used in this case.

Kyla Chasalow, Karen Levy (2021): Representativeness in Statistics, Politics, and Machine Learning. arXiv:2101.03827v3 [cs.CY] 10 Feb 2021

Heather Booth, Patrick Gerland (2015): Demographic Techniques: Data Adjustment and Correction. in: International Encyclopedia of the Social & Behavioral Sciences (Second Edition), Elsevier, 2015, Pages 126-137, ISBN 9780080970875, https://doi.org/10.1016/B978-0-08-097086-8.31011-X.

Q: Line 706: Consider using active wording. For example:

We introduced a novel, AI-based method for riverbed sediment analysis. The method uses underwater images to reconstruct spatial variations in sediment grain sizes. Trained and validated with ~15.000 underwater images collected in a section of the Danube in Hungary, we show that the method is able to map the riverbed along the vessel's route at a high spatial density of xxx samples per meter. The method does not require a scale and thus allows the distance between the camera and the riverbed to

vary. In contrast to conventional point samples of river-bed substrate, our method provides spatially continuous data, that can be further enhanced (e.g. by interpolation) to 2D maps. The method can be applied in studies where dense information about river-bed composition is required, such as riverine habitat studies, computational hydro- and morphodynamic models, or analyses of river-restoration measures.

A: Thank you for your comment! We have followed your suggestion.

**2) Other clarifications and edits**

- The first paragraph of Section 3.3 has been rephrased.

- The last paragraph of Section 2.3 has been rephrased.

- The first paragraph of Section 3.2.1 has been rephrased.

- Line 502-508 has been rephrased.

---

## Author Response (AR5)

Dear Editorial Team,

Q: The Editor asked the following correction:

Please ensure that the colour schemes used in your maps and charts allow readers with colour vision deficiencies to correctly interpret your findings. Please check your figures using the Coblis – Color Blindness Simulator (https://www.color-blindness.com/coblis-color-blindness-simulator/) and revise the colour schemes accordingly.

A: Thank you for your comment!

We have revised our figures with the recommended Simulator and implemented changes accordingly:

*Figure 2.a and 2.b have been updated with adjusted colour scheme after validation in the Simulator. They were indeed not satisfying and hard to read with colour vision deficiencies.*

*Figure 4. - 6. were readable in the simulator, hence they were not updated.*

*Figure 7. – 9. were updated during the previous minor revisions with differently shaped symbols to decrease and negate the importance of the colour schemes.*

*Figure 10. – 11. have been now updated to comply with the requirements of readability for people with colour vision deficiencies.*

*Figure 12. was readable in the simulator, hence they were not updated.*

*Figure 13. and Figure A1. were updated during the previous minor revisions with differently shaped symbols to decrease and negate the importance of the colour schemes.*

*Figure A2. was updated during the previous minor revisions with differently shaped symbols to decrease and negate the importance of the colour schemes.*

*Figure A3., B1., B2 were readable in the simulator, hence they were not updated.*

*Figure C1., C2. were updated during the previous minor revisions with differently shaped symbols to decrease and negate the importance of the colour schemes.*

*Figure C3. has been now updated to comply with the requirements of readability for people with colour vision deficiencies.*

*Figure C4., C5. were readable in the simulator, hence they were not updated.*

*Figure D1., D2. were updated during the previous minor revisions with differently shaped symbols to decrease and negate the importance of the colour schemes.*

*Figure D3. has been now updated to comply with the requirements of readability for people with colour vision deficiencies.*

*Figure D4., D5. were readable in the simulator, hence they were not updated.*

*Figure E1., E2. were updated during the previous minor revisions with differently shaped symbols to decrease and negate the importance of the colour schemes.*

*Figure E3. has been now updated to comply with the requirements of readability for people with colour vision deficiencies.*

*Figure E4., E5. were readable in the simulator, hence they were not updated.*

*Figure F1. has 27 differently coloured curves, where the main emphasis is not on differentiating each individual curve (sample) from one another, but rather on recognising the three different behaviour groups. For this, only the positioning/distribution of the curves is sufficient to understand, and this is why the colour scheme of these 27 curves did/could not serve any strong purpose actually, neither for people without colour vision deficiencies, nor for people with colour vision deficiencies.*